# Nutrigenomic regulation of sensory plasticity

**Hayeon Sung[1†], Anoumid Vaziri[1,2†], Daniel Wilinski[1], Riley KR Woerner[1], Lydia Freddolino[3,4], Monica Dus[1,2,5]***

[1]Department of Molecular, Cellular and Developmental Biology, College of Literature, Science, and the Arts, The University of Michigan, Ann Arbor, United States; [2]The Molecular, Cellular and Developmental Biology Graduate Program, The University of Michigan, Ann Arbor, United States; [3]Department of Biological Chemistry, The University of Michigan Medical School, Ann Arbor, United States; [4]Department of Computational Medicine and Bioinformatics, The University of Michigan Medical School, Ann Arbor, United States; [5]The Michigan Neuroscience Institute, Ann Arbor, United States

**Abstract** Diet profoundly influences brain physiology, but how metabolic information is transmuted into neural activity and behavior changes remains elusive. Here, we show that the metabolic enzyme O-GlcNAc Transferase (OGT) moonlights on the chromatin of the *D. melanogaster* gustatory neurons to instruct changes in chromatin accessibility and transcription that underlie sensory adaptations to a high-sugar diet. OGT works synergistically with the Mitogen Activated Kinase/Extracellular signal Regulated Kinase (MAPK/ERK) rolled and its effector stripe (also known as EGR2 or Krox20) to integrate activity information. OGT also cooperates with the epigenetic silencer Polycomb Repressive Complex 2.1 (PRC2.1) to decrease chromatin accessibility and repress transcription in the high-sugar diet. This integration of nutritional and activity information changes the taste neurons' responses to sugar and the flies' ability to sense sweetness. Our findings reveal how nutrigenomic signaling generates neural activity and behavior in response to dietary changes in the sensory neurons.

**\*For correspondence:**
mdus@umich.edu

[†]These authors contributed equally to this work

**Competing interest:** The authors declare that no competing interests exist.

## Editor's evaluation

Using *Drosophila* gustatory neurons as a model system, the authors provide important mechanistic insight into how nutrigenomic signaling encodes nutritional information into cellular changes. The authors expand previous work by showing that OGT is associated with neural chromatin at introns and transcriptional start sites and that diet-induced changes in chromatin accessibility were amplified at loci with the presence of both OGT and PRC2.1. The work also identifies Mitogen Activated Kinase as a critical mediator in this pathway. This is an elegant group of experiments revealing mechanisms for how nutrigenomic signaling triggers cellular responses to nutrients.

## Introduction

The levels and types of dietary nutrients play an essential role in cellular processes such as growth, division, and differentiation by providing fuel and biomass. However, nutrients can also affect these aspects of cell physiology by influencing, and often orchestrating, gene expression programs (*Vaziri and Dus, 2021*; *Dai et al., 2020*). These effects are mediated through nutrient-sensitive modifications to DNA, RNA, and proteins, as well as changes to the activity, binding, and localization of enzymes and signaling factors (*Huang et al., 2015*; *Katada et al., 2012*). These nutrigenomic signaling

pathways – nutrigenomics is the field that studies food–genes interactions – could explain how the food environment affects the risk of non-communicable diseases such as diabetes, cancer, and neuro-degeneration. They also hold the potential to uncover new interventions and treatments for these debilitating diseases. While the effects of nutrients on gene expression are well established, relatively little is known about the molecular mechanisms at the food–gene interface. A significant challenge of the field has been to explain how global variations in the nutrient environment lead to changes in cell physiology and behavior, especially in neuroscience. To overcome these challenges, we have developed an experimental system where the contributions of nutrients to physiology can be studied mechanistically and in vivo (*Vaziri and Dus, 2021*). Here, we use this model to characterize how changes in metabolic signaling due to diet are integrated with cellular context to diet nutrient adaptations.

Taste sensation changes depending on diet composition. In animals, the levels of bitter, sweet, and salty foods influence how these taste stimuli are perceived, with a general inverse relationship between the amount of a particular food in the diet and the responses of the sensory system to it (*May and Dus, 2021*; *Sarangi and Dus, 2021*; *Reed et al., 2020*). For example, in humans and rodents, the dietary concentration of sugars affects sweetness intensity or the electrophysiological responses of the sensory nerves to sucrose (*Wise et al., 2016*; *McCluskey et al., 2020*; *Sung et al., 2022*; *Sartor et al., 2011*; *May and Dus, 2021*). A similar phenomenon occurs in flies, where diets supplemented with 15–30% sucrose, glucose, or fructose decrease the responses of the sensory neurons to sucrose and the transmission of the sweetness signal to higher brain areas (*May et al., 2019*; *Vaziri et al., 2020*; *Wang et al., 2020*; *Ganguly et al., 2021*; *May et al., 2020*). In rats and flies, the dulling of the sensory system to sugar occurs even without weight gain, suggesting that diet exposure is sufficient to drive sweet-taste plasticity (*Sung et al., 2022*; *May et al., 2019*). Our previous work in flies implicated metabolic signaling through the Hexosamine Biosynthesis Pathway (HBP) enzyme O-GlcNAc Transferase (OGT) in this phenomenon (*May et al., 2019*). Specifically, knockdown of *OGT* exclusively in the fly sweet-taste cells prevented the neural and behavioral decrease in sugar responses observed with a high-sugar diet (*May et al., 2019*). OGT uses the metabolic end-product of the HBP, UDP-GlcNAc, to post-translationally modify proteins and change their stability or activity (*Hart, 2019*). OGT activity is sensitive to all cellular levels of UDP-GlcNAc without substrate inhibition, but it is enhanced by high dietary sugar due to a higher flux through the HBP (*Hanover et al., 2010*; *Bouché et al., 2004*; *Hawkins et al., 1997*; *Marshall et al., 2004*; *Wang et al., 1998*; *Olivier-Van Stichelen et al., 2017*; *May et al., 2019*; *Wilinski et al., 2019*; *Na et al., 2015*). OGT is also a nucleocytoplasmic protein that interacts with many chromatin- and DNA-modifying complexes; as such, it is thought to function as a nutrigenomic sensor, bridging diet and genes (*Olivier-Van Stichelen et al., 2017*; *Olivier-Van Stichelen and Hanover, 2015*; *Hart, 2019*; *Hardivillé and Hart, 2014*). Despite global changes in HBP flux with high dietary sugar, the consequences of OGT activity differ among cell types. Understanding how this occurs would provide an opportunity to study how nutrigenomic signaling is integrated with cell-specific contexts, like activity, to generate unique adaptations. Here, we exploited the effects of OGT on *Drosophila* sensory neurons and the exquisite genetic tools of this organism to investigate this question. Our experiments reveal that nutrigenomic signaling synergizes metabolic state with ongoing cellular physiology to integrate cellular signals. In the sensory neurons, OGT decorates nutrient-sensitive loci also occupied by the epigenetic silencer PRC2.1 and the activity-dependent ERK effector Stripe (Sr). This cooperation leads to changes in chromatin accessibility and transcription that drive sensory plasticity, and the catalytic activity of OGT plays an instructional role in this process. Thus, our results uncover mechanistic insights into how nutrigenomic signaling translates nutritional information into dietary adaptations in the sensory neurons.

## Results
### The nutrient sensor OGT decorates the chromatin of sweet sensory cells

Since transcriptional changes have been implicated in sugar diet-induced taste plasticity (*Vaziri et al., 2020*; *May et al., 2019*; *Wang et al., 2020*) and OGT associated with chromatin-binding factors (*Vella et al., 2013*; *Gao et al., 2018*; *Hart et al., 2011*; *Gambetta and Müller, 2015*), we asked whether this metabolic enzyme moonlights on the chromatin of sweet-taste neurons. We used <u>D</u>NA <u>a</u>denosine <u>m</u>ethyltransferase <u>Id</u>entification (Dam-ID or TaDA) to measure the association of OGT

with DNA (*Marshall et al., 2016*; *van Steensel and Henikoff, 2000*) and Chromatin Accessibility profiling using Targeted DamID (CaTaDA) to assess chromatin accessibility (*Sen et al., 2019*). Transgenic *UAS-LT3-Dam::OGT* or *UAS-LT3-Dam* flies were crossed with *Gustatory Receptor 5*a GAL4 (*Gr5a*) flies (*Chyb et al., 2003*) to drive expression exclusively in the ~60 sweet-taste cells of the fly mouthpart, and *Tubulin-GAL80ts* to control the timing of transgene induction. *Gr5a>LT3-Dam::OGT; tubulin-GAL80ts* (green) and *Gr5a>LT3-Dam; tubulin-GAL80ts* (yellow) transgenic flies were kept at the permissive temperature and fed a control (CD, 5% sucrose) or sugar (SD, 30% sucrose) diet for 3 days (*Figure 1A*). *Dam::OGT* and *Dam* were then induced by heat shocking the animals at 28°C for 18 hr on day 4, as in our prior experimental design (*Figure 1A*; *Vaziri et al., 2020*). The normalized *Dam::OGT* replicates clustered together by diet (*Figure 1—figure supplement 1A*), and the chromatin accessibility profile of Dam at the *Gr5a* sweet-taste receptor gene promoter was high, while at the bitter *Gustatory Receptor 66a (Gr66a)* promoter – only expressed in adjacent cells – accessibility was low (*Figure 1—figure supplement 1B*), suggesting that these transgenes were targeted to the correct cells.

Dam::OGT was associated with chromatin at introns (51%) and transcriptional start sites (TSSs) and promoters (30%) (*Figure 1B*, *Supplementary file 1*; all peaks); these patterns are similar to those observed in the only other study that measured OGT occupancy on the chromatin of mouse embryonic stem cells (*Vella et al., 2013*). In flies, chromatin has been classified into five types according to the histone modifications present and the proteins bound (*Filion et al., 2010*). In this chromatin characterization, there are three types of heterochromatin, developmentally regulated 'black' chromatin (high in Histone (H) 3 Lysine (K) methylation, H3K27m), Heterochromatin-protein 1 (HP1) associated 'green' chromatin (high in H3K9me2), and Polycomb group proteins-bound 'blue' chromatin, and two types of euchromatin (high in H3K4me2 and H3K79m3), the actively elongating H3K36me3 'yellow' chromatin enriched in nucleic acid metabolism genes and the 'red' chromatin enriched in other cellular processes (*Filion et al., 2010*). Our analysis found that OGT was enriched in transcriptionally active yellow euchromatin (453 genes), consistent with its role in splicing, and at 'blue' Polycomb heterochromatin (415 genes), consistent with the known associations between Polycomb Group proteins and this metabolic enzyme (*Gambetta et al., 2009*; *Hart, 2019*; *Figure 1C*). As expected, the accessibility at yellow chromatin intervals was higher than that of blue chromatin regions (*Figure 1—figure supplement 1C*).

We next examined the differential binding of OGT between the two diets. Although the majority of intervals were shared between a CD and SD (*Figure 1E*, find_peaks False Discovery Rete (FDR) <0.01), a few hundred loci were uniquely associated with OGT in either the CD (36%) or SD (10%) only conditions. However, the chromatin accessibility at OGT-bound peaks decreased in the high-sugar diet condition (*Figure 1F*, both at blue and yellow regions, *Figure 1—figure supplement 1C*). To characterize the function of the genes occupied by OGT, we performed pathway enrichment analysis using iPAGE (*Goodarzi et al., 2009*). On CD only, OGT-decorated genes were involved in signal transduction, membrane potential, and calmodulin-dependent protein kinase activity (*Figure 1G*, left). Instead, genes targeted by OGT in the SD-only condition were enriched in G-protein-coupled receptor activity, synaptic target attraction, and transcription (*Figure 1G*, left). Finally, genes with differential OGT binding between SD/CD were enriched for regulatory/signaling and neural GO terms, including dendrite morphogenesis, neuron projection membrane, synaptic target attraction, signal transduction, pattern formation, and asymmetric cell division (*Figure 1G* right, for full iPAGE, GO term analysis see *Figure 1—figure supplement 2*). Interestingly, when we examined the pathways associated with genes found in OGT-associated blue and yellow chromatin intervals, only the blue genes revealed strongly significant enrichment in GO terms. These blue Polycomb chromatin genes were involved in GO terms such as dendrite morphogenesis (8.9E−08), axon guidance (7.88E−04), actin filament organization (2.45E−07), transcription factor activity (3.8E−07), MAPK kinase signaling (9.18E−05), and Transforming Growth Factor β pathway (0.0042). In contrast, the yellow genes only showed a small enrichment for plasma membrane, transcription factor activity (2.9E−01), basolateral plasma membrane (2.4E−01), and phosphonate metabolism (2.0E−01). Together, these experiments show that OGT resides on the chromatin of the sweet taste at open domains characterized by a small but significant diet sensitivity; genes associated with neural functions are abundant among the set with diet-dependent OGT binding but only enriched in the blue H3K27 Polycomb chromatin.

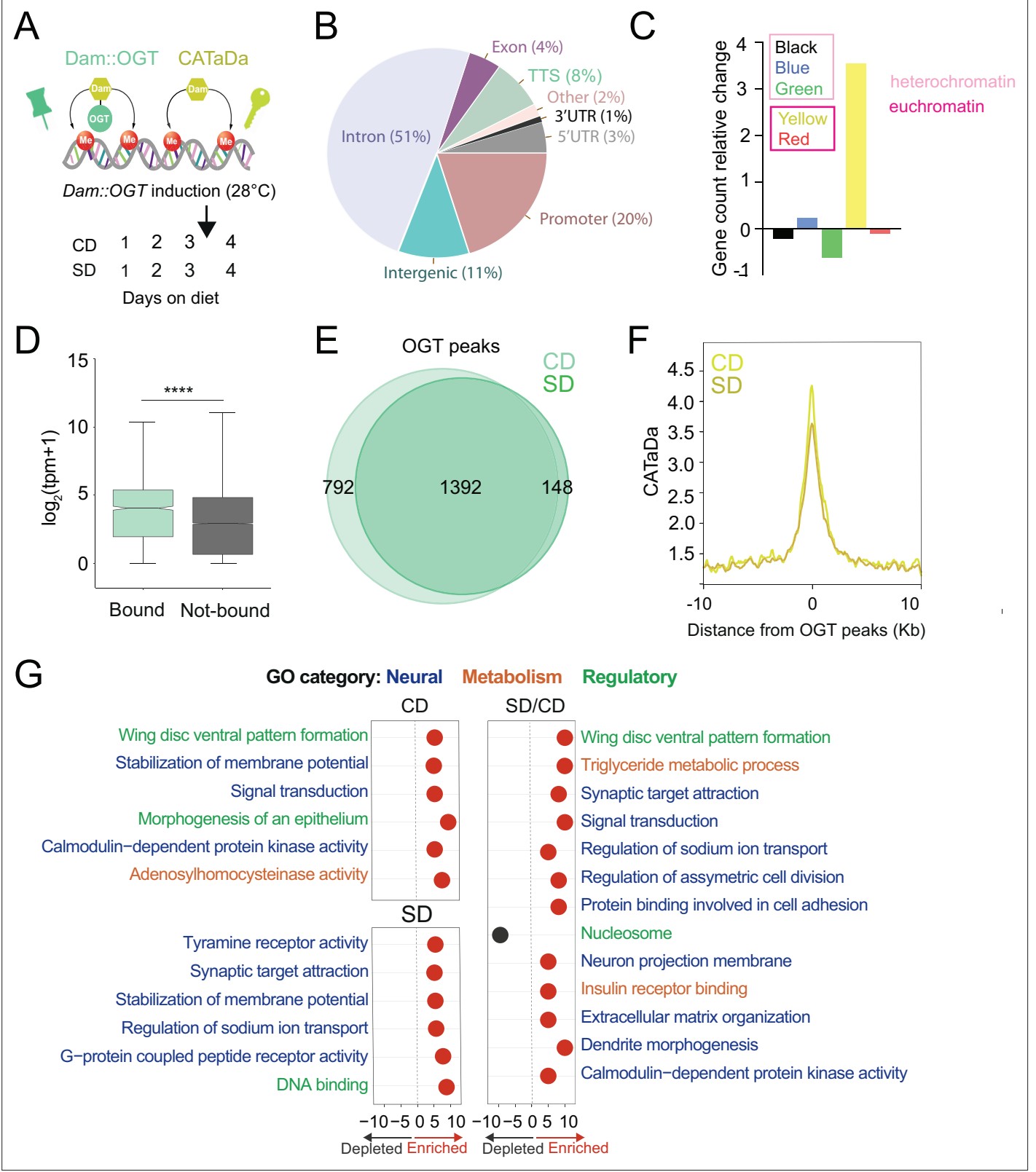

**Figure 1.** O-GlcNAc Transferase (OGT) decorates the chromatin of the sweet-taste cells. (**A**) Design of Targeted Dam-ID for OGT occupancy (*Dam::OGT*) and Dam accessibility (CATaDa) experiments. Age-matched *Gr5a;tubulin-GAL80ᵗˢ>UAS-LT3-Dam::OGT* and *Gr5a;tubulin-GAL80ᵗˢ>UAS-LT3-Dam* flies were placed on a CD or SD for 3 days at 20–21°C and then switched to 28°C between days 3 and 4 to induce expression of the transgenes. (**B**) Annotation of OGT chromatin occupied regions (all peaks) using HOMER. (**C**) The proportion of observed Dam::OGT consensus peaks allocated

*Figure 1 continued on next page*

*Figure 1 continued*

to their respective chromatin domains normalized to the expected proportions across the whole genome. Heterochromatin: black, high in H3K27m; green, bound by HP-1, high in H3K9me2; blue, bound by Polycomb Group Proteins, high in H3K27m. Yellow and red euchromatin are high in H3K4me2 and H3K79m3; yellow is also enriched for H3K36me3. (**D**) The distribution in normalized reads (Transcript Per Million, TPM + 1) for genes occupied by OGT (green). Two-tailed t test, ****p<0.0001 (**E**) Overlap of log2(Dam::OGT/Dam) chromatin-binding peaks of CD (light green) and SD (dark green) (find_peaks, q < 0.01). (**F**) Average CATaDa signal on CD (light yellow) and SD (dark yellow) centered at OGT peaks. (**G**) iPAGE summary plots for OGT peaks on a CD (top left), SD (bottom left), and the difference of SD/CD (right). Text in blue represents neural GO terms, orange represents metabolic GO terms, and green represents regulatory GO terms.

The online version of this article includes the following figure supplement(s) for figure 1:

**Figure supplement 1.** O-GlcNAc Transferase (OGT) resides on the chromatin of the Gr5a+ neurons.

**Figure supplement 2.** Pathway enrichment analysis of O-GlcNAc Transferase (OGT) chromatin targets in the Gr5a+ neurons.

## OGT and PRC2.1 share diet-sensitive chromatin sites

Our previous work showed that the epigenetic silencer PRC2.1 – specifically its H3K27m activity – was necessary and sufficient to drive sweet-taste plasticity in response to the nutrient environment (*Vaziri et al., 2020*). In the presence of high dietary sugar, PRC2.1 decreased chromatin accessibility and expression of transcription factors involved in synaptic function and signaling; these genes were located in blue-Polycomb H3K27m chromatin. Silencing these genes and their regulons lowered neural and behavioral responses to sweetness in high-sugar diet flies (*Vaziri et al., 2020*). Since OGT and PRC2.1 play a role in sweet-taste plasticity and OGT occupancy was enriched at blue Polycomb chromatin for neural functions, we asked whether there was an overlap in their occupancy.

A comparison of the peaks occupied by both Dam::Pcl (pink, Pcl is the recruiter for PRC2.1) and Dam::OGT (green) revealed a small number of shared intervals (*Figure 2A*, ~10%; *Supplementary file 1*). These 162 loci were enriched in the blue 'Polycomb' chromatin (p < 0.001, permutation test) and had lower expression levels in the Gr5a+ neurons (from TRAP experiment in *Vaziri et al., 2020*) compared to those bound by OGT alone (*Figure 2B*, *purple* vs. *green*), which include both Polycomb 'blue' and actively transcribed 'yellow' chromatin regions (*Figure 1C*; *Filion et al., 2010*). OGT × Pcl intervals had higher expression than those occupied by PRC2.1 alone, suggesting they could represent a subtype of Polycomb blue chromatin (*Figure 2B*, *purple* vs. *pink*). We next asked whether the dietary environment changed the association of OGT and Pcl at these loci. There was more OGT and Pcl at the OGT × Pcl shared sites in the SD condition compared to CD, and more OGT than Pcl was present at these sites in both diets (*Figure 2C*). Strikingly, chromatin accessibility at OGT × Pcl was markedly (50%) decreased on SD compared to CD (*Figure 2D*). This nutrient-dependent shift in accessibility was threefold higher at the shared loci compared to those bound by OGT alone (compare *Figures 1F and 2D*; also comparatively higher than those bound by PRC2.1 alone, *Vaziri et al., 2020*).

GO term analysis of genes shared by OGT/Pcl targets revealed enrichment in regulatory pathways involved in sequence-specific DNA binding, including those implicated in neural differentiation, sodium channel regulator activity, Transforming Growth Factor β and Activin receptor signaling, and dendrite development (*Figure 2E*, left). 30% of the OGT × Pcl sites corresponded to genes encoding DNA-binding and regulatory factors, including two Homeobox transcription factors known to play a role in sweet-taste function and plasticity, cad and Ptx1 (*Figure 2E*, right) (*Vaziri et al., 2020*). Analysis of protein interactions between OGT × PRC2.1 genes (*Szklarczyk et al., 2020*) uncovered a Protein–Protein Interaction network enrichment (p < 1.0e−16) among DNA-binding factors (pink, p = 2.08e−09), Mitogen-Activated Protein Kinase (MAPK, blue, p = 0.00059), signal transduction (Transforming Growth Factor, TGF-β/Activin signaling, yellow), neuron projection (red outline, p = 4.95e−7), and response to stimuli (p = 7.15e−0.5). Consistent with OGT/Pcl targets being ~40% of OGT-associated peaks, the GO terms for the shared intervals were a subset of those enriched in the OGT-bound blue chromatin.

## The catalytic activity of OGT is required for diet-induced taste plasticity

Our data show that OGT occupies the chromatin of the sensory neurons and that its binding is diet dependent at loci also bound by PRC2.1. To understand more about the mechanisms of OGT function and, thus, nutrigenomic signaling, we examined the role of OGT activity on taste plasticity using the Proboscis Extension Response (PER). As shown in *Figure 3A*, the fly proboscis houses the cell bodies

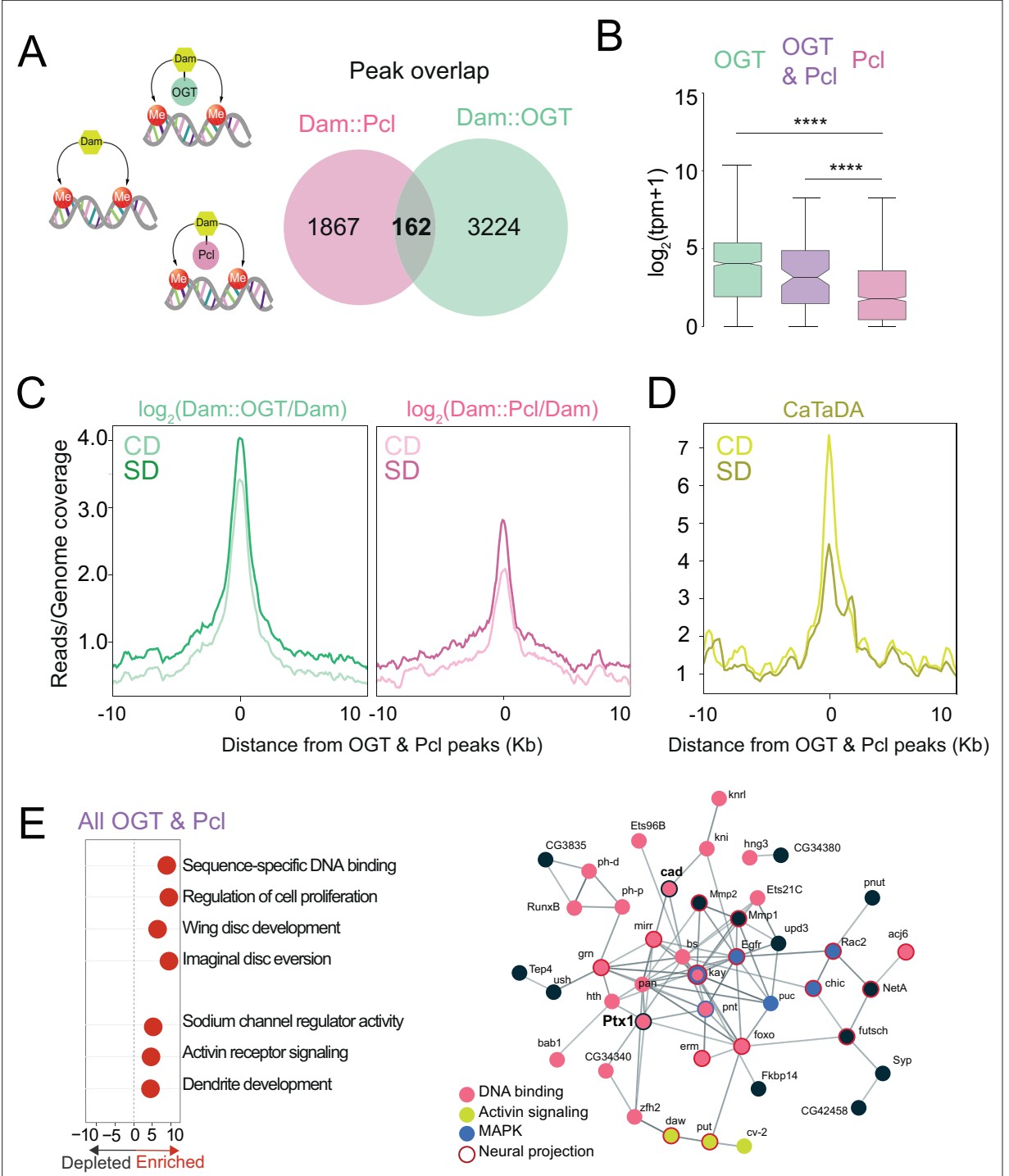

**Figure 2.** O-GlcNAc Transferase (OGT) and Polycomb Repressive Complex 2.1 (PRC2.1) mark nutrient-sensitive chromatin in the sweet-taste cells. (**A**) Diagram of the Targeted Dam-ID (TaDa occupancy, Dam::OGT green, and Dam::Pcl pink) and (CATaDA, accessibility, yellow) experiments analyzed in this figure. Overlap of log2(Dam::Pcl/Dam, pink) and log2(Dam::OGT/Dam, green) chromatin occupancy peaks (all peaks, peak calling: find_peaks, q<0.01). (**B**) The distribution in normalized reads (Transcript Per Million, TPM +1) for genes occupied by OGT (green), Pcl (pink), and OGT and Pcl (purple). Two-tailed t test, ****p<0.0001. (**C**) Average log2(Dam::OGT/Dam; left) and log2(Dam::Pcl/Dam) (right) signal on a CD (lighter shades) and SD (darker shades) diet centered at OGT + Pcl co-occupied peaks. (**D**) Average CATaDa signal on CD (lighter shade) and SD (darker shade) centered at OGT + Pcl co-occupied peaks. (**E**) (*left*) iPAGE pathway analysis of genes co-occupied by OGT and Pcl and (*right*) STRING interaction network of genes co-occupied by OGT + Pcl, colors represent GO terms from the pathway enrichment analysis.

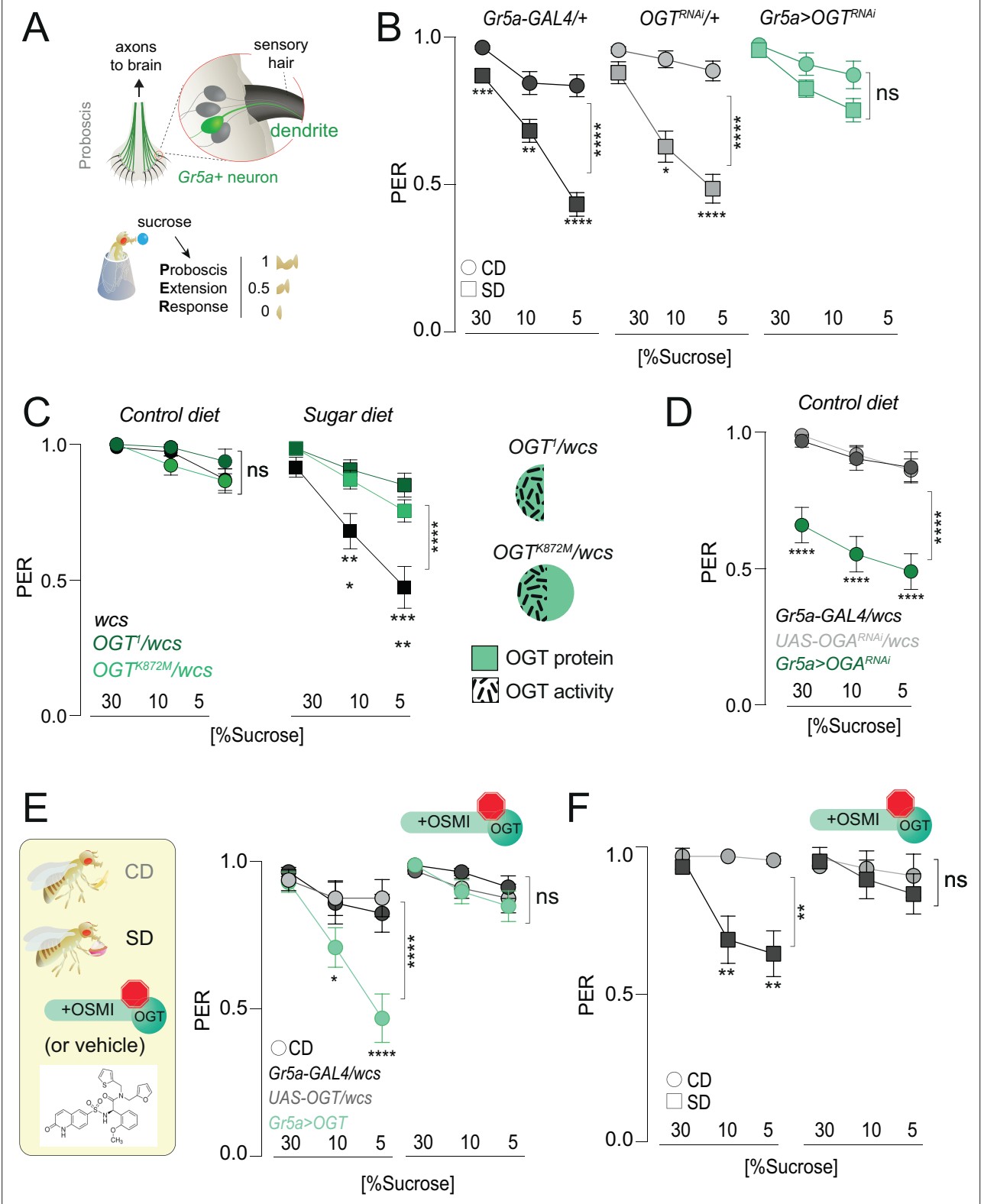

**Figure 3.** O-GlcNAc Transferase (OGT) activity is necessary for taste plasticity in response to the sugar diet environment. (**A**) (*top*) Anatomy of the sensory system showing the cell bodies, dendrites, and axons of the sweet-sensing Gr5a+ neurons; (*bottom*) Diagram of the Proboscis Extension Response (PER). (**B**) Taste responses (*y*-axis) to stimulation of the labellum with 30, 10, and 5% sucrose (*x*-axis) in flies with knockdown of *OGT* (green) or controls (shades of gray) in flies fed a CD (circles) or SD (squares); *n* = 18–51. Two-way repeated measure analysis of variance (ANOVA), main effect

*Figure 3 continued on next page*

Figure 3 continued

of genotype: *Gr5a>wcs* p < 0.0001 (Tukey multiple comparison 30% p = 0.0008, 10% p = 0.0047, 5% p < 0.0001), *Gr5a>OGT-RNAi* p = 0.2657 (Sidak multiple comparison 30% p = 0.2792, 10% p = 0.9756, 5% p = 0.4883), *OGT-RNAi>wcs* (Sidak multiple comparison 30% p = 0.5923, 10% p = 0.0381, 5% p < 0.0001). (**C**) Taste responses (*y*-axis) to stimulation of the labellum with 30, 10, and 5% sucrose (*x*-axis) in flies with mutations in *OGT* (green) or controls (black) while on a control diet (CD, n = 16–26) or SD, n = 26–29. Two-way repeated measure ANOVA, main effect of genotype compared to *wcs* controls: ****p < 0.0001. Tukey multiple comparisons test, ****p < 0.0001, ***p < 0.001, **p < 0.01, *p < 0.05. (*right*) Diagram of *OGT* alleles: $OGT^1$, protein null, $OGT^{K872M}$ catalytically dead. (**D**) Taste responses (*y*-axis) to stimulation of the labellum with 30, 10, and 5% sucrose (*x*-axis) in flies with knowdown of *OGA* (green) or controls (shades of gray) while on a control diet (CD), n = 31–47. Two-way repeated measure ANOVA, main effect of $Gr5a>OGA^{RNAi}$ genotype compared to each control genotype: ****p < 0.0001. Tukey multiple comparisons test, ****p < 0.0001. (**E**) (*left*) Diagram of experiments with 10 µM OGT Small Molecule Inhibitor-1 (OSMI-1) in E and F; (*right*) taste responses (*y*-axis) to stimulation of the labellum with 30, 10, and 5% sucrose (*x*-axis) in flies with overexpression of *OGT* (green) or controls (shades of gray) while on a control diet (CD) supplemented with OSMI or vehicle, n = 19–22. Two-way repeated measure ANOVA, main effect of *Gr5a>OGT* genotype compared to each control genotype: ****p < 0.0001. Tukey multiple comparisons test, ****p < 0.0001 and *p < 0.05. (**F**) Taste responses (*y*-axis) to stimulation of the labellum with 30, 10, and 5% sucrose (*x*-axis) of age-matched male $w1118^{cs}$ flies on a CD (circle) or SD (square) diet with vehicle (DMSO) or OSMI. n = 14–17. Two-way repeated measure ANOVA, main effect of OSMI treatment p = 0.0089; Tukey multiple comparisons test for 30, 10, and 5% sucrose: (1) CD vs. SD (DMSO) p < 0.05, p = 0.0090, p = 0.0034 and (2) CD vs. SD (+OSMI-1), p > 0.05 at all concentrations. Data are shown as mean ± standard error of the mean (SEM).

and dendrites of the sensory neurons. When the taste sensilla in the labellum are stimulated with sucrose, the fly extends its proboscis to reach the sweet solution. The amount of proboscis extension for each concentration tested – 1 is a full extension, 0.5 a half, and 0 none – corresponds to the fly's ability to taste and can be compared across genotypes and diets. As previously shown, consumption of SD for 7 days results in a decrease in PER for high (30%) and low (5%) concentrations of sucrose compared to animals that ate a control diet (*Figure 3B*, circles vs. squares, gray shades). However, knocking down *OGT* in the Gr5a+ sweet-sensing neurons resulted in flies with similar sweet sensitivity between the two diets (*Figure 3B*). Thus, OGT is required for diet-dependent sweet-taste plasticity. To ask if the catalytic activity of OGT was required for this taste phenotype, we compared the ability of protein null ($OGT^1$) and catalytically dead mutants ($OGT^{K872M}$) to rescue taste plasticity; both of these alleles are homozygous lethal and thus were tested in combination with *w1118CS* control flies. Neither mutant affected sweet-taste responses on a control diet, but both prevented the lower PER to sucrose observed in SD-fed control flies (*Figure 3C*). This argues that the catalytic activity of OGT is required for the effects of this enzyme on taste plasticity. Consistent with this, knocking down the antagonistic enzyme *O-GlcNAcase* (*OGA*), which removes the GlcNAc moiety from proteins, resulted in lower sweet-taste responses on CD (*Figure 3D*). We next asked if increasing the levels of OGT was sufficient to induce sweet-taste changes. OGT activity is linear across all levels of cellular UDP-GlcNAc, so increasing its levels also increases its activity (*Hart, 2019*). Overexpression of *OGT* in the Gr5a+ neurons resulted in sucrose responses on CD comparable to those observed in sugar diet-fed flies (*Figure 3E*). However, inhibiting the activity of OGT with the specific OGT Small Molecule Inhibitor-1 (OSMI) (*Ortiz-Meoz et al., 2015*; *May et al., 2020*) blocked the effects of OGT overexpression on sweet-taste responses (*Figure 3E*, right); this drug treatment had no effect on survival (*Figure 4—figure supplement 1A*). To finally test if the effects of SD on taste plasticity were dependent on OGT activity, we supplemented the CD and SD with OSMI during the entire duration of the diet exposure (7 days) and then tested PER to sucrose. Control (vehicle, Dimethyl Sulfoxide (DMSO)) flies exhibited a dulling of sweet-taste responses on SD (squares), but this decrease was entirely blocked by OSMI (*Figure 3F*); no effects were observed on a CD (circles). Thus, decreasing the activity of OGT, either with genetics or pharmacological tools, resulted in similar effects on taste plasticity, arguing that the activity of this enzyme plays an essential role in taste changes in response to the dietary environment.

## OGT and PRC2.1 genetically interact to drive taste plasticity

To determine the effects of OGT catalytic activity on chromatin accessibility and PRC2.1 occupancy, we fed *Gr5a>LT3-Dam; tubulin-GAL80ts* and *Gr5a>LT3-Dam::Pcl; tubulin-Gal80ts* (*Figure 4—figure supplement 1A*) flies a CD or an SD supplemented with OSMI. Strikingly, OSMI treatment completely abolished the changes in chromatin accessibility observed with SD at OGT × Pcl sites (*Figure 4A*, compared to *Figure 2D*; *Supplementary file 1*), suggesting that OGT activity is necessary for diet-dependent dynamics at these loci. However, inhibition of OGT activity did not affect Pcl occupancy at these peaks, indicating that recruitment of PRC2.1 to these sites is largely independent of this metabolic enzyme (pink, *Figure 4A*). OSMI also had a mild effect on the occupancy of Dam::Pcl

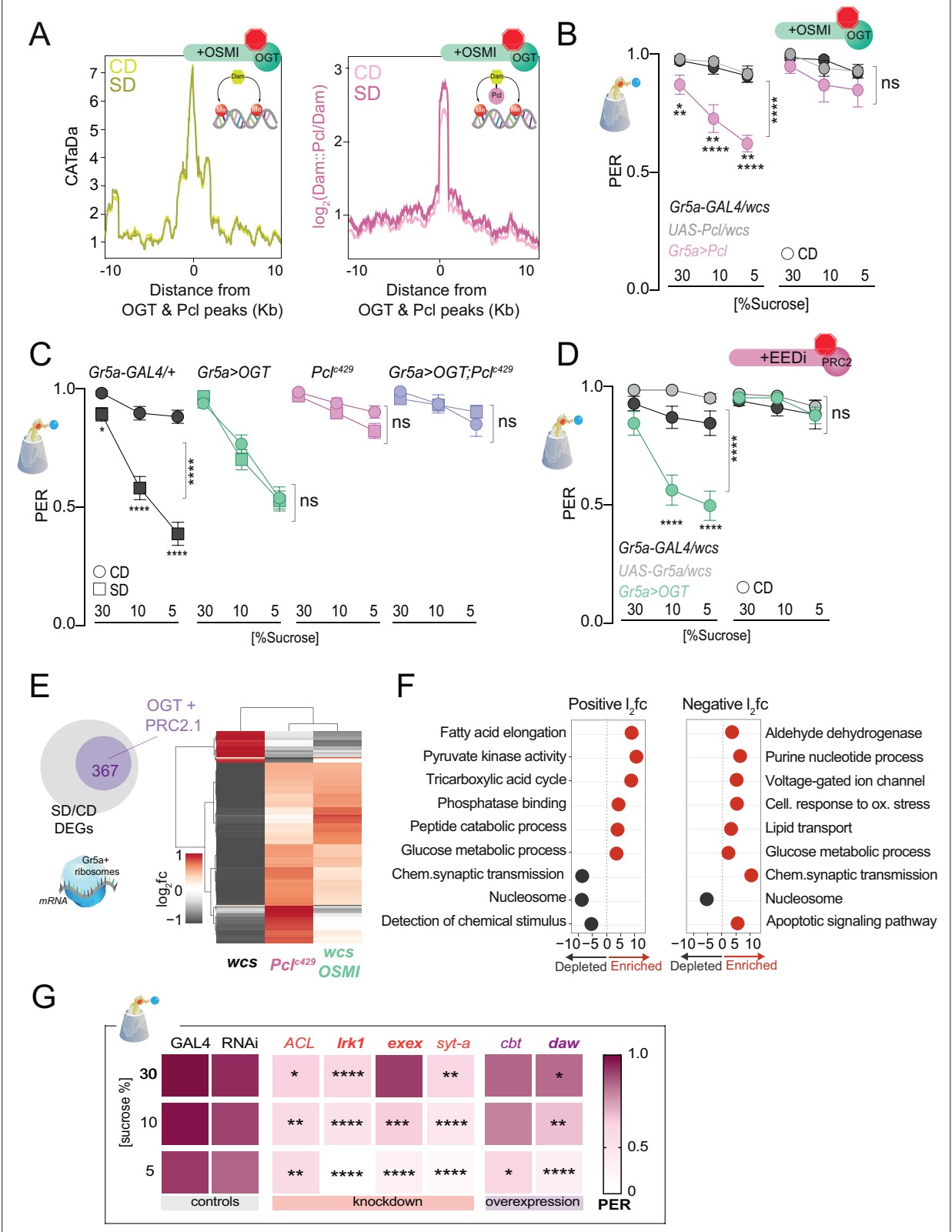

**Figure 4.** O-GlcNAc Transferase (OGT) activity is necessary for chromatin and transcriptional dynamics in response to the sugar diet environment. (**A**) (*left*) Average CATaDa signal on a CD (light yellow) and SD (dark yellow) with OSMI centered at OGT + Pcl peaks (compare to *Figure 2D*); (right) Average log2 Dam::Pcl/Dam signal on a CD (light pink) and SD (dark pink) with OSMI centered at OGT + Pcl peaks (compare to *Figure 2*). (**B**) Taste responses (*y-axis*) to stimulation of the labellum with 30, 10, and 5% sucrose (*x-axis*) of age-matched male *Gr5a>Pcl* (pink) and transgenic controls

*Figure 4 continued on next page*

*Figure 4 continued*

(shades of gray) on CD supplemented with vehicle (DMSO, *n* = 14–23) or 10 µM OSMI (*n* = 20–42). Two-way repeated measure analysis of variance (ANOVA): (1) DMSO, main effect of genotype (****p < 0.0001) and genotype x concentration (*p < 0.05); Tukey multiple comparisons test for 30, 10, and 5% sucrose concentrations: *Gr5a>wcs* vs. *Gr5a>Pcl* p = 0.0165, p = 0.0056, p = 0.0025; *Pcl>wcs* vs. *Gr5a>wcs*, ns. (2) OSMI: main effect of genotype p = 0.3194 and genotype × concentration p = 0.6893. (**C**) Taste responses (*y*-axis) to stimulation of the labellum with 30, 10, and 5% sucrose (*x*-axis) in *Gr5a>OGT;Pcl^{c429}* (purple), *Gr5a>OGT* (green), *Pcl^{c429}/+* (pink), and transgenic controls (gray) on CD or SD, *n* = 20–37. Two way repeated measure ANOVA, main effect of diet: ****p < 0.0001. Tukey multiple comparisons test, ****p < 0.0001 and *p < 0.05. (**D**) Taste responses (*y*-axis) to stimulation of the labellum with 30, 10, and 5% sucrose (*x*-axis) in *Gr5a>OGT* (green) and transgenic controls (shades of gray) on a CD supplemented with vehicle or 8 µM EEDi, *n* = 20–22. Two-way repeated measure ANOVA, main effect of *Gr5a>OGT* genotype compared to each control genotype: ****p < 0.0001. Tukey multiple comparisons test, ****p < 0.0001. (**E**) Log$_2$fold (l2fc) of differentially expressed genes (DEGs) between SD/CD in *w1118cs* ± OSMI and *Pcl^{c429}* SD/CD. (**F**) GO term analysis of the DEGs measured in the Gr5a+ neurons of flies fed a CD and SD + OSMI. (**G**) Taste responses (*y*-axis) to stimulation of the labellum with 30, 10, and 5% sucrose (*x*-axis) for a subset of DEGs in (E, purple circle) that show dependence on OGT and Polycomb Repressive Complex 2.1 (PRC2.1). *n* = 14–49. Purple, knockdown; red, overexpression; bold, direct OGT/PRC2.1 targets. Two-way repeated measure ANOVA, main effect of *GAL4>wcs* control genotype compared to each control genotype: ****p <0.0001. Tukey multiple comparisons test, ****p < 0.0001, ***p < 0.001, **p < 0.01, *p < 0.05. ACL, ATP Citrate Lyase; Irk1, inwardly rectifier potassium channel 1; daw, dawdle; cbt, cabut; exex, extra extra; syt-a, synaptotagmin alpha. Data are shown as mean ± standard error of the mean (SEM).

The online version of this article includes the following figure supplement(s) for figure 4:

**Figure supplement 1.** Pcl occupancy at PRE and Pcl peaks with inhibition of O-GlcNAc Transferase (OGT) activity.

**Figure supplement 2.** Pathway enrichment analysis of the effects of OSMI on Dam::Pcl peaks.

**Figure supplement 3.** Transcriptional responses to the dietary environment when O-GlcNAc Transferase (OGT) activity was inhibited.

**Figure supplement 4.** Pathway enrichment analysis of genes reverted or unchanged by OSMI.

genomewide since the number (~1800), and identity (80%) of Dam::Pcl peaks were mainly the same with or without OSMI (***Figure 4—figure supplement 1C***). Only a smaller fraction of new PRC2.1-only peaks emerged with OSMI treatment, and the genes in these intervals were enriched in GO terms such as detection of chemical stimuli, DNA binding, and protein kinase activation (***Figure 4—figure supplement 2***). Thus, OGT activity is required for the diet-dependent decrease in chromatin accessibility but not PRC2.1 recruitment or occupancy, suggesting that other factors or events mediate these. However, we found that the catalytic activity of OGT was necessary for PRC2.1-mediated taste modulation. Overexpression of *Pcl* in the Gr5a+ neurons mimics the effects of SD on taste by decreasing responses to sucrose in flies fed a CD (***Figure 4B***, *left*) – a result dependent on the H3K27 methylation activity of this complex (***Vaziri et al., 2020***). However, OSMI blocked the effects of *Pcl* overexpression on sucrose responses compared to vehicle-fed flies (***Figure 4B***, *right*). These results place OGT upstream of PRC2.1 at both the molecular and behavioral levels. Since OGT is an enzyme known to modify proteins, we also asked whether its effects on taste plasticity were mediated, at least in part, through PRC2.1. *Pcl^{c429}* mutations blocked diet-induced taste plasticity on SD (***Figure 4C***, pink vs. black, compare squares vs. circles), while *Gr5a>OGT* overexpression promoted a decrease in sucrose PER even on CD (***Figure 4C***, green vs. black, compare circles and squares). However, when *Pcl^{c429}* mutants were combined with *Gr5a>OGT*, these flies failed to develop taste plasticity in response to SD without any effects on CD (***Figure 4C*** purple). Thus, *Gr5a>OGT;Pcl^{c429}* phenocopied *Pcl^{c429}* mutants (compare pink and purple), suggesting that the effects of OGT act largely through PRC2.1. To further confirm these results and link them to the catalytic H3K27 methylation activity of PRC2.1, we treated *Gr5a>OGT* and control flies with a vehicle or the specific inhibitor of PRC2 (EEDi) while on CD (***Qi et al., 2017***). This manipulation restored normal taste responses to control levels in *Gr5a>OGT* flies (***Figure 4D***, compare right vs. left), consistent with what we observed with *Gr5a>OGT; Pcl^{c429}* flies (***Figure 4C***). Together these results argue for a strong genetic interaction between OGT and PRC2.1.

To understand the consequences of the observed OGT-dependent shifts in chromatin accessibility, we isolated mRNAs associated with the ribosomes of the *Gr5a+* cells using Translating mRNA Affinity Purification (TRAP) (***Chen and Dickman, 2017***) in flies fed a CD + OSMI and SD + OSMI. Principal component analysis revealed that most of the variation between samples was due to diet (***Figure 4—figure supplement 3A***); mRNAs specifically expressed in the *Gr5a+* cells, such as the sweet-taste receptor genes (*Gr5a*, *Gr64f*, and *Gr64a)* and the fatty acids taste receptor *Ir56D*, were enriched in the *Gr5a+* fraction compared to the input, while bitter receptor genes (*Gr66a* and *Gr32a)* were depleted (***Figure 4—figure supplement 3B***), indicating that the selection of *Gr5a+* mRNAs was successful and comparable to our prior experiments (***Vaziri et al., 2020***). However, compared to the marked

negative skew in gene expression we previously observed with a high-sugar diet, where 90% of genes had negative log2 fold changes, OSMI-differentially expressed genes (DEGs) showed a similar distribution in positive and negative changes (*Figure 4—figure supplement 3C*; *Supplementary file 1*; *Vaziri et al., 2020*). Indeed, further analyses revealed that OSMI treatment *reverted* (i.e., showed the opposite direction of change; $q < 0.1$, Wald test) or *restored* (practical equivalence test using a null hypothesis of a change of at least 1.5-fold and $q < 0.05$) the expression of 52% of the DEGs with SD/CD change (*Figure 4E*, gray are downregulated and red are upregulated), and that most of the genes changed with OSMI treatment (367) were also similarly affected by a loss of function *Pcl^{c429}* mutation (*Figure 4E*). These genes were enriched in metabolic and neural processes, such as chemical synapse transmission, synaptic target attraction, cell differentiation, glucose metabolism, and detection of chemical stimuli (*Figure 4G* and *Figure 4—figure supplement 4*). Notable among these were the homeobox *cad* and *Ptx1* and their regulons, which have been implicated in taste function and plasticity (*Figure 4—figure supplement 3D*; *Vaziri et al., 2020*), but also many whose effects of sweet-taste sensation are not known, like the Activin ligand *dawdle* (*daw*) and the transcription factor *cabut* (*cbt*). To functionally validate some of these new genes, we knocked down or overexpressed them (depending on their SD/CD log2 fold) in the Gr5a+ neurons and assayed sweet-taste function by PER (*Figure 4F*). This revealed a mild-to-moderate effect of these genes on sucrose sensation; interestingly, *daw* and *Irk1* are direct targets of Pcl and OGT.

Together, these experiments place OGT and PRC2.1 in the same genetic pathway that directs diet-induced taste plasticity at the chromatin, transcriptional, and behavioral levels. They also argue that the activity of the metabolic sensor OGT may provide the nutrient-dependent context for PRC2.1-mediated changes in chromatin accessibility (but not PRC2.1 recruitment), while PRC2.1 may instead function as an effector of these chromatin and transcriptional changes.

## The transcription factor Sr is part of the OGT–PRC2.1 nutrigenomic pathway

Our data show that OGT orchestrates responses to the dietary environment in the sensory neurons. To understand which other cellular context factors may cooperate with OGT and PRC2.1 to mediate taste plasticity, we examined the regulatory regions of OGT × Pcl and OGT × PREs (Polycomb Responsive Elements, DNA motifs to which Polycomb Proteins bind) loci for enriched cis-regulatory motifs (*Figure 5—figure supplement 1A, B* and *Supplementary file 1*; also, note that a TF may appear multiple times as dots in this graph because of different binding motifs). In this analysis, the highest log2 fold enrichments were for Polycomb Group proteins like *PhoRC* (1.018, $p = 0.0099$) and Trx-recruiter *Trithorax-like* (*Trl*, 1.972, $p = 0.0099$ and 0.803, $p = 0.0099$; *Trx* is antagonistic to PRC2.1), as well as transcription factors, such as the Zn-finger immediate early gene *Stripe* (*Sr*, homolog of activity-dependent human Early Growth Response 2, EGR2, alias Krox20), and the nutrient-sensitive factor *Sterol-Responsive-Element Binding Protein* (*SREBP*) (*Supplementary file 1*). To determine if they affected sweet taste, we measured the proboscis extension in response to sucrose when these genes were overexpressed or knocked down in the Gr5a+ neurons (*Figure 5—figure supplement 1C*). The only factor that affected sweet-taste responses across low and high sucrose concentrations was Sr, which also showed higher mRNA abundance in the sensory neurons of SD-flies (*Figure 5—figure supplement 1D*).

Sr is a conserved transcription factor induced by neural activity via the MAPK/ERK pathway (*Chen et al., 2016*; *Gonzales et al., 2020*; *Beckmann and Wilce, 1997*) and is essential for sensory nerve development and plasticity (*Murphy et al., 1989*; *Duclot and Kabbaj, 2017*). The MAPK pathway is sensitive to neural activity and nutrients (*Robles-Flores et al., 2021*; *Papa et al., 2019*); it is stimulated by mitogens, such as TGF-β/Activin signaling, which increase with high-sugar levels, eating, and neural activity in flies and mammals (*Lavoie et al., 2020*; *Liu and Chen, 2022*; *Wilinski et al., 2019*). Of note, OGT × Pcl co-occupied loci were enriched in MAPK/ERK targets (*Figure 2E*, blue). Enrichment for *Sr* cis-regulatory motifs was modest at OGT *or* Pcl-only loci (l2fc = 0.196 and 0.398, respectively; $p < 0.001$ in both cases via an approximate permutation test) but strong at genes bound by both factors (OGT × Pcl = 0.429 and OGT × PREs = 0.976, $p < 0.001$, permutation test) (*Figure 5A*; *Supplementary file 1*). When we compared the distribution of Sr sites around the regulatory regions of OGT × PRC2.1 genes (either together or separately), we found a marked bias around the TSS (*Figure 5B*, top), with enrichment of Sr-binding sites in the 500 bp window immediately preceding

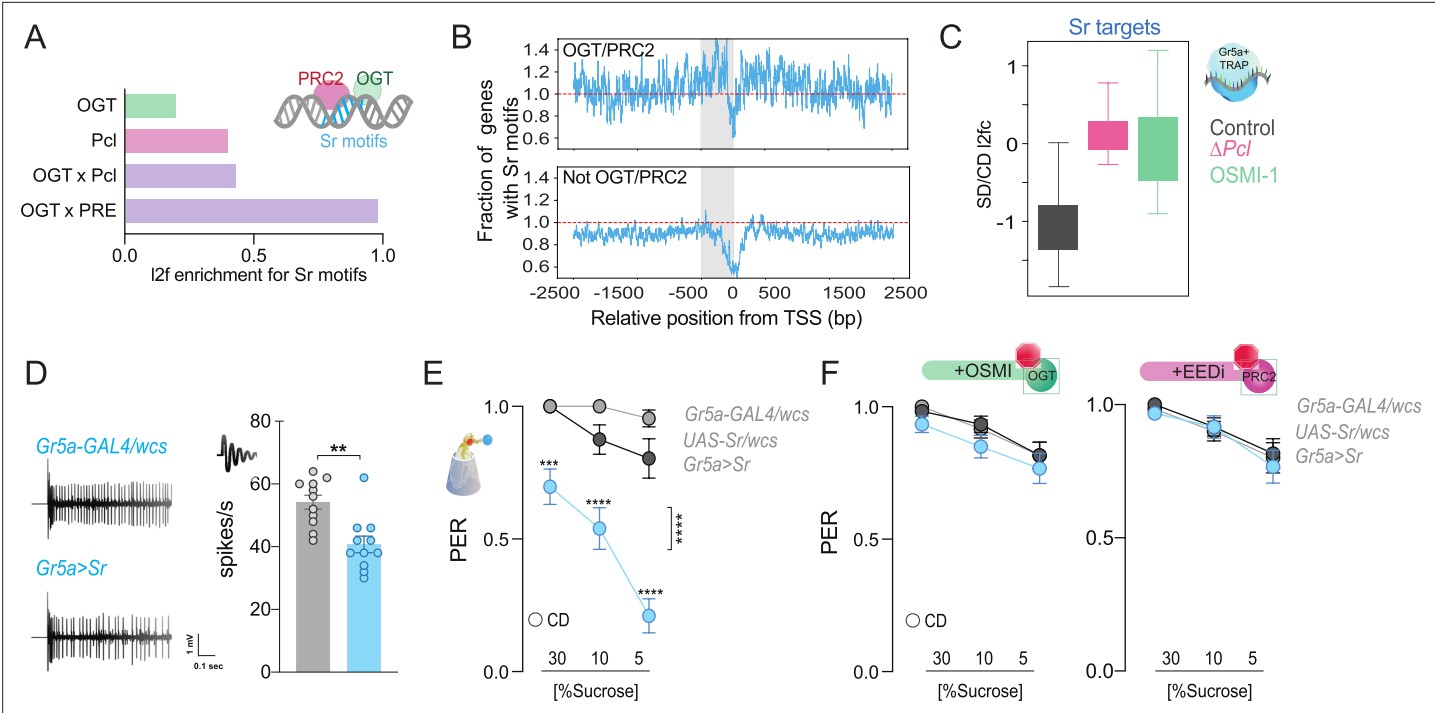

**Figure 5.** The immediate early gene Sr is found at PRC2 × OGT genes and is involved in sweet-taste sensation. (**A**) Log$_2$fold (l2f) enrichment for Sr motifs at sites occupied by O-GlcNAc Transferase (OGT; green), Pcl (pink), or OGT + Pcl and OGT + PREs (purple), p < 0.0001. (**B**) Normalized distribution of Sr motifs along the regulatory regions 2500 bp up and downstream the transcriptional start site (TSS) for the genes in (**A**); the 500 bp immediately before each TSS are shaded in grey. Counts are normalized relative to the genome-wide expectation (derived by multiplying the number of potential target genes by the fractional coverage of Sr motifs on the genome); a score of 1.0 indicates the hypothetical genome-wide average overlap with Sr motifs and is shown as a red dashed line. (**C**) The distribution of RNA l2fc for genes that have Sr sites and are expressed in the Gr5a+ neurons of flies on a CD and SD in control flies or flies with mutations in Pcl or inhibition of OGT. q > 0.01. (**D**) Representative traces (left) and averaged neuronal responses to 25 mM sucrose stimulation from the L-type sensilla of Sr overexpression flies (blue) and controls (gray). n = 11. Mann–Whitney test: **p = 0.001. (**E**) Taste responses (y-axis) to stimulation of the labellum with 30, 10, and 5% sucrose (x-axis) in Sr overexpression flies (blue) and controls (gray). n = 22–38. Two-way repeated measure analysis of variance (ANOVA), main effect of genotype ****p < 0.0001 and genotype × concentration ****p < 0.0001; Tukey post-test for multiple comparisons: 30%: ***p = 0.0002 for Gr5a>Sr compared to each control, 10%: Gr5a>Sr vs. Gr5a>wcs p = 0.0025 and Gr5a>Sr vs. Sr >wcs ****p < 0.0001; 5%: ****p < 0.0001 for Gr5a>Sr compared to each control. Gr5a>wcs vs. Sr>wcs p > 0.05 at all concentrations. (**F**) Taste responses (y-axis) to stimulation of the labellum with 30, 10, and 5% sucrose (x-axis) in Sr overexpression flies (blue) and controls (gray) treated with the OGT inhibitor OSMI (green) or the PRC2 inhibitor EEDi (pink). n = 30. Two-way repeated measure ANOVA, main effect of genotype p = 0.2993 and p = 0.9146 and genotype × concentration p = 0.9293 and p = 0.9146, respectively. Data are shown as mean ± standard error of the mean (SEM).

The online version of this article includes the following figure supplement(s) for figure 5:

**Figure supplement 1.** Enrichment analysis of cis-regulatory sites present in O-GlcNAc Transferase (OGT) and PREs.

the TSS. In contrast, genes that OGT or PRC2.1 did not occupy, were depleted for Sr-binding sites at the TSS (*Figure 5B*, bottom). This suggests that Sr is enriched at the TSS of genes that are bound by OGT *and* PRC2.1. To this end, when we examined the expression of Sr-targets in the Gr5a+ neurons of flies on the two diets, we noticed that these genes had negative log2 fold changes on the sugar diet; this repression, however, was abolished by mutations in *Pcl* and by inhibition of OGT activity (*Figure 5C*, compare gray vs. pink and green, respectively), which hints toward functional cooperation between Sr and OGT/PRC2.1. To characterize the effects of higher *Sr* levels on neural activity and behavior, we used the UAS/GAL4 system to overexpress this gene in the Gr5a+ neurons of adult flies. Overexpression of *Sr* resulted in lower electrophysiological responses of the gustatory neurons to sucrose (*Figure 5D*, Mann–Whitney test, *p* = 0.001) as well as lower PER at both high and low sucrose concentrations (*Figure 5E*). This effect, however, was dependent on the catalytic activities of PRC2 and OGT. Indeed, overexpression of *Sr* in flies treated with OSMI (green) or with an inhibitor of PRC2 (pink, EEDi *Vaziri et al., 2020*) resulted in sucrose responses comparable to those of control flies (*Figure 5F*). Overall, these results argue for a role of Sr in taste plasticity in coordination with OGT and PRC2.1. Importantly, since OGT and PRC2.1 are not always found at TSS, the bias in Sr distribution at

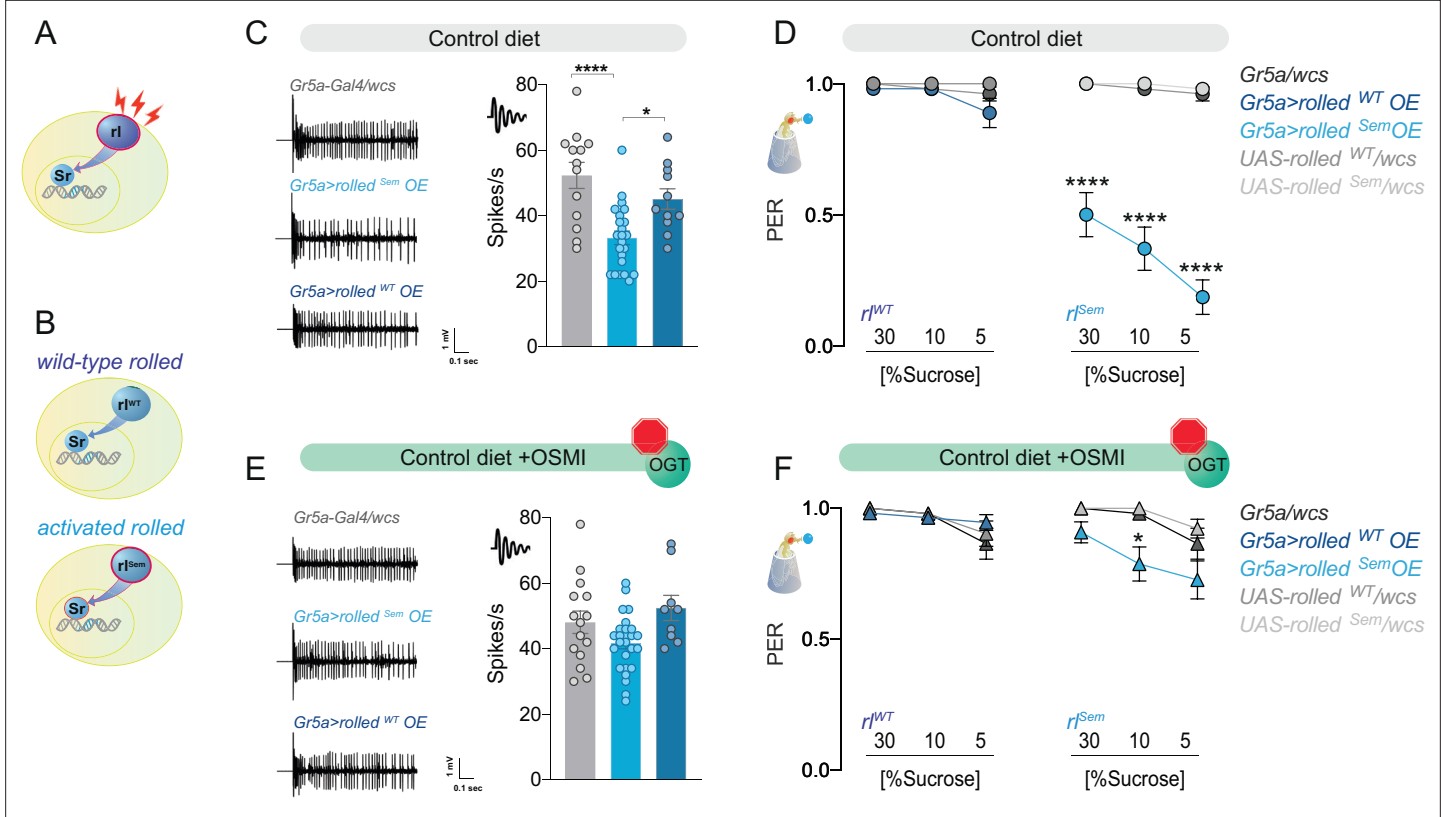

**Figure 6.** The effect of the kinase rl/ERK on sweet taste depends on O-GlcNAc Transferase (OGT) activity. (**A**) Diagram of the rl/ERK > Sr pathway, red sparks represent neural activity, and red outline represents activation. (**B**) Diagram of the two types of *rl/ERK* transgenes used. (**C**) Representative traces (left) and averaged neuronal responses to 25 mM sucrose of L-type sensilla in overexpression of wild-type ($rl^{WT}$) or constitutively active ($rl^{Sem}$) rl/ERK in the Gr5a+ neurons (blue) and control (gray) on a CD. $n$ = 11–23. One-way analysis of variance (ANOVA); Tukey's multiple comparison test: ****$p < 0.0001$ for *Gr5a/+* vs. *Gr5a>rl^{Sem}*, $p$ = 0.279 for *Gr5a/+* vs. *Gr5a>rl^{WT}*, and *$p$ = 0.018 for *Gr5a>rl^{Sem}* vs. *Gr5a>rl^{WT}*. (**D**) Taste responses (y-axis) to stimulation of the labellum with 30, 10, and 5% sucrose (x-axis) in flies with overexpression of wild-type ($rl^{WT}$) or constitutively active ($rl^{Sem}$) rl/ERK in the Gr5a+neurons (blue) and control (gray) flies on a CD + vehicle (DMSO). $n$ = 24–27. Two-way repeated measure ANOVA, main effect of genotype $p < 0.0001$ and concentration × genotype $p < 0.0001$. Tukey multiple comparisons tests: ****$p < 0.0001$ for *Gr5a>rl^{Sem}* vs. all other genotypes at 30, 10, and 5% and $p > 0.05$ for all other comparisons at all concentrations. (**E**) Representative traces (left) and averaged neuronal responses to 25 mM sucrose of L-type sensilla in overexpression of wild-type ($rl^{WT}$) or constitutively active ($rl^{Sem}$) rl/ERK in the Gr5a+ neurons (blue) and control (gray) on a CD + OSMI. $n$ = 11–23. One-way ANOVA; Tukey's multiple comparison test: $p$ = 0.172 for Gr5a/+ vs. *Gr5a>rl^{Sem}*, $p$ = 0.603 for Gr5a/+ vs. *Gr5a>rl^{WT}*, and $p$ = 0.034 for *Gr5a>rl^{Sem}* vs. *Gr5a>rl^{WT}*. (**F**) Taste responses (y-axis) to stimulation of the labellum with 30, 10, and 5% sucrose (x-axis) in flies with overexpression of wild-type ($rl^{WT}$) or constitutively active ($rl^{Sem}$) rl/ERK in the Gr5a+ neurons (blue) and control (gray) in flies fed a CD + OSMI. $n$ = 26–33. Two-way repeated measure ANOVA, main effect of genotype $p$ = 0.005; Tukey multiple comparisons tests: $p > 0.05$ for all other comparisons at all concentrations except for $p < 0.0001$ for *Gr5a>rl^{Sem}* vs. $rl^{Sem}$/wcs at 10% $p$ = 0.0216. Effect of OSMI vs. vehicle: *Gr5a>rl^{Sem}* 30% $p$ = 0.0012, 10% $p$ = 0.0030, 5% $p < 0.0001$, and $p < 0.05$ for all other genotypes. Data are shown as mean ± standard error of the mean (SEM).

The online version of this article includes the following figure supplement(s) for figure 6:

**Figure supplement 1.** rl/ERK is required for taste responses to sugar.

the TSS of 'OGT/PRC2.1 regulated genes' indicates that coordination between Sr and OGT/PRC2.1 may arise not from direct interactions but rather from two distinct paths of information flow, such as metabolism and neural activity.

## The ERK pathway modulates taste adaptations in response to diet

Sr is the downstream transcriptional effector for the Extracellular-signal Regulated Kinase (ERK), a pathway stimulated by neural activity that plays a role in plasticity (*Lavoie et al., 2020*; *Miningou and Blackwell, 2020*; *Thomas and Huganir, 2004*; *Figure 6A*). We reasoned that ERK/EGR2 might provide sensory neurons with a specific context to drive dietary adaptations. To test this hypothesis,

we examined the role of the kinase rolled (rl) – the ERK homolog in *D. melanogaster* – in sweet-taste and diet-induced taste plasticity.

First, we observed that, as with *Sr*, the mRNA abundance of *rl* was higher in the Gr5a+ neurons of flies on SD, but this gene was not a direct target of OGT or PRC2.1 (*Figure 6—figure supplement 1A*). We tried several available antibodies against rl and activated (phosphorylated) rl to establish whether increased transcript levels also resulted in higher activation of this kinase; however, none of them resulted in a reliable signal in our hands. We thus turned to genetic tools to investigate whether higher rl expression or activity played a role in sweet-taste plasticity. To differentiate between these two possibilities, we expressed either a wild-type *rl* ($rl^{WT}$, *Figure 6B*, top, *Biggs et al., 1994*) or constitutively active form of the kinase ($rl^{Sem}$ *Figure 6B*, bottom, *Oellers and Hafen, 1996*) in the Gr5a+ neurons and tested neural and taste responses to sucrose. Overexpression of $rl^{WT}$ with Gr5a-GAL4 did not affect the electrophysiological responses of the sensory neurons to sucrose (*Figure 6C*, *dark blue*); however, expression of the active $rl^{Sem}$ decreased neuronal responses to sucrose (*Figure 6C*, *light blue*). These activity phenotypes were reflected in the behavioral taste responses to sucrose, with $rl^{WT}$ flies having identical PER to sucrose as controls and $rl^{Sem}$ showing reduced PER across high and low sucrose concentrations (*Figure 6D*, *left* vs. *right*; note that controls are shared here, plotted separately for clarity). Thus, rl activity, but not higher levels alone, was sufficient to affect sweet-taste plasticity. Not surprisingly, given the known function of rl/ERK in neural activity, we found that this kinase was also necessary for sweet-taste responses, as loss of function mutants and RNAi for *rl* had lower electrophysiological and behavioral responses to sucrose (*Figure 6—figure supplement 1B, C*). To test whether there was a synergetic interaction between rl and OGT, we repeated the same experiments in the presence of the OGT inhibitor OSMI. Strikingly, OSMI treatment almost entirely blocked the effects of $rl^{Sem}$ on both neural and behavioral responses to sucrose while having no effect on $rl^{WT}$ flies (*Figure 6E, F*).

To characterize the function of the rl/Sr pathway on taste plasticity, we used Trametinib, a drug that inhibits ERK activation in animals (currently used for treating melanoma). At concentrations previously used in flies (15.6 µM) (*Castillo-Quan et al., 2019*; *Slack et al., 2015*), Trametinib blocked the effects of $rl^{Sem}$ expression on sweet-taste responses (*Figure 7—figure supplement 1A*); this had no effect on survival (*Figure 4—figure supplement 1A*). Treatment with this ERK inhibitor also negated the effects of *Sr* overexpression on sucrose responses, resulting in flies with PER comparable to controls (*Figure 7—figure supplement 1B*), placing Sr downstream of ERK activation. Thus, Trametinib treatment efficiently blocks ERK signaling. To determine if the activity of the rl/Sr (ERK/EGR2) pathway was necessary for taste plasticity in response to the sugar diet environment, we fed flies a control or sugar diet with or without Trametinib for 7 days, then measured their neural and behavioral responses to sucrose. Exposure to a high-sugar diet decreased the electrophysiological (*Figure 7A*) and behavioral (*Figure 7B*) responses to sucrose. However, when rl activity was blocked with Trametinib, there was no decrease in neural responses or PER (*Figure 7C and D*). Of note, Trametinib had a minor but significant effect on sweet-taste activity (compare CD of *Figure 7C* with CD of *Figure 7A*), consistent with the observation that rl is necessary for normal sweet-taste function (*Figure 6—figure supplement 1B, C*). Together, these data indicate that the ERK pathway plays a critical role in the development of taste adaptations in response to diet and place its function upstream of OGT.

## Discussion

Nutrigenomic signaling plays a role in health and disease by bridging the dietary environment with physiological adaptations. However, the molecular mechanisms and consequences of this type of nutrient sensing are still poorly understood. In particular, how nutrigenomic signals are integrated with cellular contexts has remained hard to define due to the lack of mechanistic nutrigenomic models (*Müller and Kersten, 2003*; *Vaziri and Dus, 2021*). In this work, we exploited the conserved phenomenon of diet-induced taste plasticity and the genetic tools of the *D. melanogaster* fly to answer these questions.

Here, we report that the metabolic enzyme OGT is associated with neural chromatin at introns and TSSs. While OGT-associated genes showed subtle but significant changes in chromatin accessibility in response to diet, these dynamics were much more robust at loci co-occupied by both OGT and the epigenetic silencer PRC2.1. At genes decorated by both factors, we observed sizable diet-dependent chromatin variations that were critically dependent on the catalytic activity of OGT. OGT

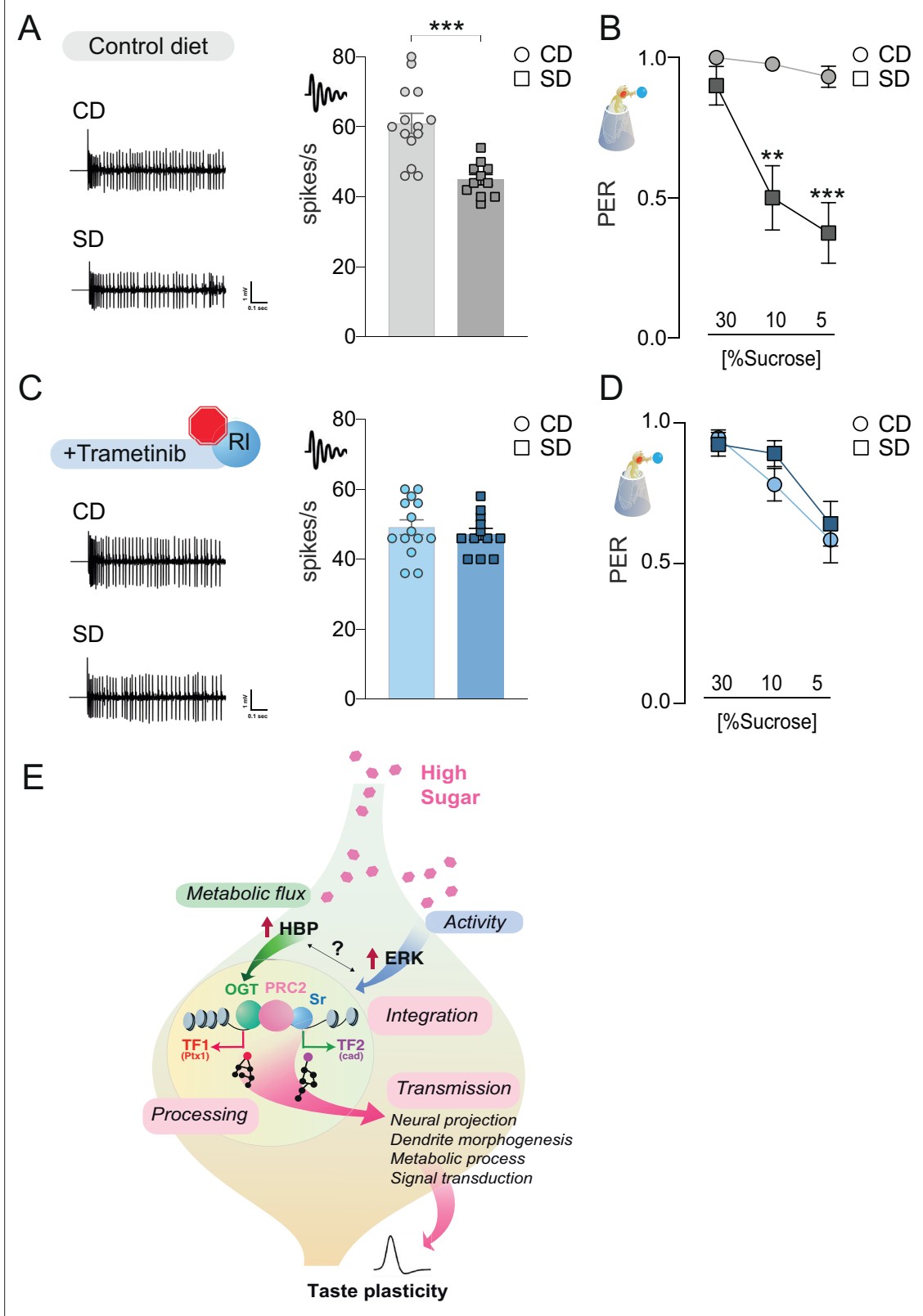

**Figure 7.** The rl > Sr pathway is important for taste adaptations in response to diet. Representative traces (left) and averaged responses to 25 mM sucrose from L-type sensilla of flies fed a CD and SD (**A**), gray or Trametinib (**C**), blue. *n* = 11–14. Unpaired *t*-test: ***p = 0.0001 for CD vs. SD, and p = 0.486 for CD Trametinib vs. SD Trametinib. Taste responses to stimulation of the proboscis with sucrose in flies fed a CD and SD + vehicle (**B**), DMSO, gray or Trametinib (**D**), blue. PER, *n* = 20–33. PER: two-way repeated measure analysis of variance (ANOVA), main effect of diet, vehicle p < 0.0001 and

*Figure 7 continued on next page*

*Figure 7 continued*

Trametinib p = 0.4701; Tukey multiple comparison test: vehicle CD vs. SD 30% p = 0.421, 10% **p = 0.0017, and 5% ***p = 0.0002 and Trametinib CD vs. SD 30% p = 0.9702, 10% p = 0.4470, and 5% p = 0.9575. Effect of Trametinib: CD vehicle vs. CD Trametinib, 30% p = 0.1745, 10% p = 0.0108, 5% p = 0.0015; SD vehicle vs. SD Trametinib, 30% p = 0.4837, 10% p = 0.0228, 5% p = 0.2339. (**E**) A model for how O-GlcNAc Transferase (OGT), PRC2, and ERK orchestrate taste plasticity in response to a changing food environment. Boxes in pink describe the different steps of 'information processing' (see discussion). Data are shown as mean ± standard error of the mean (SEM).

The online version of this article includes the following figure supplement(s) for figure 7:

**Figure supplement 1.** Effects of ERK inhibitor Trametinib on genetic manipulations of *Sr* and *rl* levels.

activity was also necessary for the differential transcriptional and taste responses to the high-sugar diet. The OGT- and PRC2.1-bound nutrient-dependent loci were enriched for binding motifs for the activity-dependent transcription factor Sr at TSS, the effector of the ERK pathway. We show that this signaling pathway functions upstream of OGT/PRC2.1 to shape neural and behavioral taste responses to the dietary environment. We thus propose a model where a nutrigenomic pathway composed of OGT and PRC2.1 integrates information from the nutrient and cellular environment via ERK signaling to orchestrate sensory responses to diet (*Figure 7E*). Our data suggest that this integration occurs at the level of chromatin and modulates the expression of transcription factors and signaling regulators that further amplify and extend the reach of nutrigenomic signaling. Our findings thus shed light on how nutrigenomics contributes to neural plasticity and behavior.

## OGT and chromatin dynamics

OGT is a conserved enzyme that catalyzes the transfer of UDP-GlcNAc to the serine and threonine residues of proteins (*Hart, 2019*; *Olivier-Van Stichelen et al., 2017*). Because UDP-GlcNAc synthesis by the HBP combines sugar, amino acid, nucleotide, and fatty acid metabolism, the levels of this metabolite, as well as the activity of the enzymes in this pathway, are inextricably linked to cellular metabolism and diet (*Hart, 2019*; *Olivier-Van Stichelen et al., 2017*). Higher HBP flux directly impacts OGT activity because the function of this enzyme is linear across a vast range of physiological UDP-GlcNAc concentrations. Because of this, OGT is recognized as a critical nutrient sensor in animal physiology, particularly in development, cancer, and metabolic disease (*Hart, 2019*; *Olivier-Van Stichelen et al., 2017*). More recently, its importance for neural function and plasticity has also been recognized, with studies implicating it in synapse maturation, neural excitability, activity, and plasticity (*Butler et al., 2019*; *Ardiel et al., 2018*; *Lagerlöf et al., 2016*; *Su and Schwarz, 2017*; *Giles et al., 2019*; *Lagerlöf et al., 2017*; *Ruan et al., 2014*; *Hwang and Rhim, 2019*; *Li et al., 2019*; *May et al., 2019*). Our group showed that a high-sugar diet acutely and chronically increased HPB activity in flies and played a role in diet-induced sensory plasticity (*Wilinski et al., 2019*; *May et al., 2019*).

OGT is a nucleocytoplasmic enzyme, and the GlcNAc modification is enriched in nuclear and synaptic proteins, which extends its reach on cellular physiology (*Hart, 2019*; *Olivier-Van Stichelen et al., 2017*). Although OGT is thought to play a role in gene regulation, only one study has shown its direct association with chromatin in murine embryonic stem cells (*Vella et al., 2013*). Here, we report that OGT also decorates neural chromatin in *Drosophila melanogaster*. Like in murine embryonic stem cells, OGT was enriched at introns and TSSs and primarily associated with transcriptionally active chromatin. However, we found that half of OGT intervals were also enriched at Polycomb repressive chromatin, consistent with previous reports that the GlcNAc modification is found at PREs, as well as on many Polycomb Group (PcG) proteins; OGT is also associated with PcG complexes to mediate Hox-gene repression (*Hart, 2019*; *Olivier-Van Stichelen et al., 2017*; *Schuettengruber et al., 2017*). On a high-sugar diet, there was a higher association of OGT with DNA but lower chromatin accessibility. However, the magnitude of these changes depended on what other regulatory and DNA-binding factors were found at OGT loci. At loci with PRC2.1 binding and Sr/EGR2 motifs, chromatin openness was markedly reduced in response to the high-sugar diet environment. This is the first study to show that OGT-associated chromatin is nutrient sensitive to the best of our knowledge. Importantly, this nutrient sensitivity was entirely dependent on the catalytic activity of OGT because it was abolished in the presence of the inhibitor OSMI. Interestingly, OGT activity had no effect on PRC2.1 association with co-occupied loci (and only a small effect at non-OGT loci, data not shown). The H3K27 methylation activity of PRC2.1 is required for changes in chromatin accessibility, including those that depend on diet in the sensory neurons (*Schuettengruber et al., 2017*; *Vaziri et al., 2020*). Thus, our data

suggest that OGT activity affects the repressive action but not the recruitment of PRC2.1; we also demonstrate that the catalytic activity of PRC2.1 is required for the effects of OGT on taste plasticity.

These findings raise several important questions about the biochemical mechanisms of OGT action that our genetic system is poorly suited to address, but that will be important to define in future studies. First, what are the targets of OGT at nutrient-sensitive loci? Is OGT directly GlcNAcylating PRC2.1 to modify its repressive drive? Several studies have linked OGT activity with the stability, chromatin occupancy, or catalytic function of Polycomb Group Proteins (*Hart, 2019*; *Olivier-Van Stichelen et al., 2017*; *Chu et al., 2014*; *Sakabe and Hart, 2010*; *Forma et al., 2018*; *Sui et al., 2020*; *Jiang et al., 2019*; *You et al., 2021*; *Decourcelle et al., 2020*; *Gambetta and Müller, 2014*). Thus, converging evidence suggests that OGT impacts different aspects of PRC2.1 and PcG biology and is broadly consistent with our data. Connections between OGT and ERK have also been uncovered in the context of cancer and cell division, with studies showing that inhibition of ERK signaling decreases O-GlcNAcylation and vice versa (*Zhang et al., 2015*; *Jiang et al., 2016*) and that GlcNAcylation promotes ERK effects while OGT inhibition blocks them (*Cork et al., 2018*; *Weiss et al., 2021*; *Lei et al., 2020*). These findings are consistent with the effect and direction of the genetic interactions we observed between OGT and PRC2.1 and OGT and ERK, the direction of 'information flow' within the cell (*Figure 7E*), and the effects of our genetic manipulations. OGT could also affect chromatin accessibility by GlcNAcylating histones, although the function and effects of these histone modifications are still poorly understood (*Gambetta and Müller, 2015*; *Hirosawa et al., 2018*; *Konzman et al., 2020*; *Olivier-Van Stichelen et al., 2017*). Another outstanding question is how OGT is targeted to chromatin and whether this recruitment is dynamic and related to nutrient availability. For example, are there different local pools of OGT and GlcNAc in the nucleus vs. cytoplasm (or mitochondria) where OGT has been described? Because of the lack of functional fly OGT antibodies, we could not ask this question, but it is possible that the levels of OGT in the nucleus and cytoplasm change between diets. Finding answers related to the cellular compartmentalization of this metabolic enzyme and its targets will be an essential step in understanding nutrient signaling.

## Sensors and effector mechanisms of nutrigenomic signaling in neural plasticity

In the case of the sweet-taste neurons, sugar directly activates the cells via receptor-dependent mechanisms and enhances OGT's metabolic activity. Our data support the idea that integrating these two signals at the level of chromatin – a synergy almost entirely unique to these cells – is key for developing sensory plasticity. First, binding sites for the ERK effector Sr were among the most enriched at OGT × Pcl/PRE loci, consistent with our finding that OGT/Pcl/PRE loci were enriched in MAPK signaling (*Figures 2 and 5*); second, diet-driven changes in Sr regulons depended on the activity or presence of OGT and PRC2.1, and most importantly, the effects of rl/Sr (ERK/EGR2) on taste plasticity had strong epistatic interactions with OGT. Together, these molecular and functional data support the idea that the activity-dependent ERK pathway provides a relevant cellular context (likely neural activity) for adaptations to the nutrient environment. However, these sensing mechanisms must be turned into action to be effective. This is likely the role of PRC2.1 since the effects of ERK/EGR2 and OGT manipulations were dependent on the PRC2.1 function. Of note, only a small portion of the genes occupied by PRC2.1 is sensitive to diet and OGT activity, and our data show that PRC2.1 is not binding to new loci but instead tuning the output of those it is already bound to, likely via OGT instruction.

In the model we propose, metabolic and activity sensors integrate cellular information to promote changes in gene expression. But how are these actualized into physiological (in this case, neural) adaptations that underlie behavior or disease? This is one of the central and unresolved questions in nutrigenomics. Our model's genetic and neural tractability provides a unique opportunity to get some answers. The 162 co-occupied loci identified were enriched for transcription and regulatory factors involved in cell proliferation, differentiation, signaling, and neural activity, as well as pathways implicated in neural plasticity. Many of these DNA-binding factors play essential roles during the development of sensory neurons to set their biophysical properties, such as Ptx1 and cad, but also affect adult taste plasticity. The regulons of these TFs include genes known to affect pre- and postsynaptic branching and structure, as well as synaptic physiology. Thus, our collective data indicate that this nutrigenomic pathway promotes taste adaptations, most likely by re-engaging developmental gene batteries, a mechanism that has been hypothesized to play a role in neural plasticity (*Hobert,*

*2011*; *Marder and Prinz, 2002*; *Parrish et al., 2014*; *Vaziri et al., 2020*). Whether this is a general rule of nutrigenomic signaling or something specific to these cells or neurons is yet to be determined; however, it is interesting to note that this is similar to how cancer cells exploit developmental networks for uncontrolled growth (*Faubert et al., 2020*; *DeBerardinis and Chandel, 2016*). On this note, many neural and psychiatric conditions show associations and connections with diet and metabolic states, including epilepsy, schizophrenia, bipolar, depression, Alzheimer's, and Parkinson's (*Sarangi and Dus, 2021*; *Grigolon et al., 2020*). Thus, uncovering nutrigenomic mechanisms in the brain could shed light on the etiology of these conditions and help design nutritional strategies to support people suffering from them; this is similar to how metabolic disorders like diabetes and cardiovascular disease are treated with a combination of drugs and nutrition.

### Limitations

Although using sensory plasticity and fly gustatory neurons as a model to study nutrigenomic signaling brings unique advantages, it also has significant limitations. These primarily arise from the small number of cells (60) and the in vivo nature of our model. First, we cannot probe whether OGT, PRC2.1, and Sr/EGR2 physically interact or modify each other in these cells. We also could not probe whether SD enhances the presence of OGT protein in the nucleus compared to a control diet due to the lack of functional antibodies. Thus, evidence for our model arises from the combination of cell-specific molecular, genetic, and physiological data. Second, we only inferred that the loci with Sr/EGR2 motifs integrate activity due to the well-established function of the ERK pathway in activity-dependent plasticity; future studies should address this directly and compare the effects of acute vs. chronic nutrient influx.

Further, while inhibitors have allowed us to establish critical epistatic interactions and conduct dietary manipulations while bypassing developmental effects and genetic challenges, we cannot exclude that some of these effects may be non-cell autonomous. Integrating this model with biochemical approaches that preserve the appropriate activity and nutrient context would help address these critical questions. Finally, pathways beyond OGT, ERK, and PRC2.1 may also play a role in sensory plasticity.

### Conclusions

In summary, we show that activity and nutrient-sensing mechanisms are integrated at the genomic level to promote neural adaptations to the food environment. In particular, our data reveal a central and instructional role for OGT and meaningful epistatic interactions with sensors (ERK) and effectors (PRC2.1). More generally, we put forth a model where cell and context specificity transforms 'nutritional data' – that is, variations in nutrient and metabolite levels – into nutritional information (*Floridi, 2005*), as shown in *Figure 7E* (pink boxes). This information is processed and interpreted by gene regulatory processes to make 'decisions' about responding to environmental challenges and carrying out physiological, neural, and behavioral changes. Thus, nutrigenomic mechanisms could provide a critical path for information flow in biological systems (*Shannon, 1948*; *Reinagel, 2000*; *Smith, 2000*; *Fabris, 2009*). A clear advantage could reside in their ability to amplify transient, and often minor, variations in nutrient and activity levels into strong reactions, which can be used to orchestrate responses to current *and* future environmental challenges. Future studies in this field will no doubt uncover fascinating insights about the rules of nutrigenomic communication: these discoveries will illuminate how nutrition and gene expression converge to shape cell physiology and provide us with new tools to promote wellness and diminish the burden of disease.

## Materials and methods

### Key resources table

| Reagent type (species) or resource | Designation | Source or reference | Identifiers | Additional information |
|---|---|---|---|---|
| Genetic reagent (*D. melanogaster*) | w[1118]CS | Other | | Gift from A Simon |

*Continued on next page*

*Continued*

| Reagent type (species) or resource | Designation | Source or reference | Identifiers | Additional information |
|---|---|---|---|---|
| Genetic reagent (*D. melanogaster*) | UAST-sxc(Ogt)RNAiCLb38 | PMID:24706800 | | Gift from C Lehner |
| Genetic reagent (*D. melanogaster*) | Gr5a-GAL4 | Bloomington *Drosophila* Stock Center | BDSC: 57592 | |
| Genetic reagent (*D. melanogaster*) | Tubulin-GAL80ts | Bloomington *Drosophila* Stock Center | BDSC: 7018 | |
| Genetic reagent (*D. melanogaster*) | UAS-Pcl | FlyORF | FlyORF: F001897 | |
| Genetic reagent (*D. melanogaster*) | Pcl[c429] | Other | | Gift from N Liu |
| Genetic reagent (*D. melanogaster*) | UAS-Rpl3-3XFLAG | PMID:29194454 | | Gift from D Dickman |
| Genetic reagent (*D. melanogaster*) | UAS-LT3-Dam | Other | | Gift from AH Brand |
| Genetic reagent (*D. melanogaster*) | UAS-LT3-Dam::Pcl | PMID:33177090 | | |
| Genetic reagent (*D. melanogaster*) | UAS-LT3-Dam::OGT | This paper | | See the Materials and methods |
| Genetic reagent (*D. melanogaster*) | Gr64f-GAL4 | Bloomington *Drosophila* Stock Center | BDSC: 57669 | |
| Genetic reagent (*D. melanogaster*) | UAS-Sr | Bloomington *Drosophila* Stock Center | BDSC: 26553 | |
| Genetic reagent (*D. melanogaster*) | UAS-rl[WT] | Bloomington *Drosophila* Stock Center | BDSC: 36270 | |
| Genetic reagent (*D. melanogaster*) | UAS-rl[Sem] | Bloomington *Drosophila* Stock Center | BDSC: 59006 | |
| Genetic reagent (*D. melanogaster*) | rl[1] | Bloomington *Drosophila* Stock Center | BDSC: 386 | |
| Genetic reagent (*D. melanogaster*) | rl RNAi | Bloomington *Drosophila* Stock Center | BDSC: 34855 | |
| Genetic reagent (*D. melanogaster*) | OGA RNAi | Bloomington *Drosophila* Stock Center | BDSC: 41882 | |
| Genetic reagent (*D. melanogaster*) | OGT[1] | PMID:26348912 | | Gift from D van Aalten |
| Genetic reagent (*D. melanogaster*) | OGT[K872M] | PMID:26348912 | | Gift from D van Aalten |
| Antibody | Mouse monoclonal anti-Flag | Sigma | Cat#: F1804, RRID:AB_262044 | 3:50 |
| Peptide, recombinant protein | Dynabeads Protein G | Thermo Fisher Scientific | Cat#: 10004D | |
| Peptide, recombinant protein | T4 DNA ligase | New England Biolabs | Cat#: M0202S | |
| Commercial assay or kit | NEBuilder HiFi DNA Assembly kit | New England Biolabs | Cat#: E5520S | |
| Commercial assay or kit | ThruPLEX Kit | Takara | Cat#: 022818 | |
| Chemical compound, drug | OSMI-1 | Sigma | Cat#: SML1621 | |
| Chemical compound, drug | EED226 | Axon Medchem | Cat#: 2701 | |

*Continued on next page*

*Continued*

| Reagent type (species) or resource | Designation | Source or reference | Identifiers | Additional information |
|---|---|---|---|---|
| Chemical compound, drug | Trametinib | LC labs | Cat#: T-8123 | |
| Chemical compound, drug | TRIzol LS Reagent | Thermo Fisher Scientific | Cat#: 10296010 | |
| Software, algorithm | Autospike3.9 | Syntech | | http://www.ockenfels-syntech.com/products/signal-acquisition-systems-2/ |
| Software, algorithm | Prism 9 | GraphPad | RRID:SCR_002798 | |
| Software, algorithm | Python | Python | RRID:SCR_008394 | |

## Fly husbandry, strains, and diets

All flies were grown and fed cornmeal food (Bloomington Food B recipe) at 25°C and 45–55% humidity under a 12 hr light/12 hr dark cycle (Zeitgeber time 0 at 9:00 AM) unless otherwise stated. Male flies were collected under $CO_2$ anesthesia 1–3 days after eclosion and maintained in a vial that housed 35–40 flies. Flies were acclimated to their new vial environment for an additional 2 days and were moved to fresh food vials every other day. The GAL4/UAS system was used to express transgenes of interest using the *Gustatory receptor 5a Gr5a-GAL4* transgene. For each GAL4/UAS cross, transgenic controls were made by crossing the *w1118^CS* (gift from A. Simon, *CS* and *w1118* lines from the Benzer laboratory) to GAL4 or UAS flies, sex-matched to those used in the GAL4/UAS cross. The fly lines used for this paper are listed in *Supplementary file 1*.

For all dietary manipulations, the following compounds were mixed into standard cornmeal food (Bloomington Food B recipe) (0.58 calories/g) by melting, mixing, and pouring new vials as in *Musselman and Kühnlein, 2018* and *Na et al., 2013*. For the 30% sugar diet (1.41 calories/g), Domino granulated sugar (wt/vol) was added. Inhibitors were solubilized in 10% DMSO and added to the control o sugar diet at a total concentration of 10 μM for OSMI (*Ortiz-Meoz et al., 2015*; *May et al., 2020*), 8 μM for EEDi (*Vaziri et al., 2020*), and 15.6 μM for Trametinib (*Castillo-Quan et al., 2019*; *Slack et al., 2015*). Animals were assigned randomly to dietary groups. The sample sizes were determined based on standards in the field. No animal was excluded from any of the analyses.

## Proboscis extension response

Male flies were food deprived for 18–20 hr in a vial with a Kimwipe dampened with Milli-Q filtered deionized water. PER was carried out as described in *Shiraiwa and Carlson, 2007*. Extension responses were recorded manually, and experimenters were blinded whenever possible. Experiments were replicated two to three times by two different experimenters.

## Affinity purification of ribosome-associated mRNA (TRAP)

Male fly heads (300 per replicate, ~10,000 *Gr5a+* cells) were collected using sieves chilled in liquid nitrogen and dry ice. Frozen tissue was then lysed as previously described (*Chen and Dickman, 2017*; *Vaziri et al., 2020*). From 10% of the total lysate, total RNA was extracted by TRIzol LS Reagent (Thermo Fisher Scientific, 10296010) for input. The remainder of the lysate was precleared by incubation with Dynabeads Protein G (Thermo Fisher Scientific, 10004D) for 2 hr and subsequently incubated with Dynabeads Protein G and an anti-Flag antibody (Sigma-Aldrich, F1804) at 4°C with rotation for 2 hr, then RNA was extracted from ribosomes bound to beads by TRIzol Reagent (*Chen and Dickman, 2017*).

## Targeted DNA adenine methyltransferase identification (TaDa) and chromatin accessibility TaDa (CATada)

To generate the *UAS-LT3-Dam::OGT* construct, the coding region of the *OGT* gene was amplified from *w1118^CS* animals with the primers listed below and assembled into the *UAS-LT3-DAM* plasmid (gift from A. Brand, University of Cambridge) using the NEBuilder HiFi DNA Assembly kit based on the manufacturer's instructions (New England Biolabs). Transgenic animals were validated by reverse transcription PCR targeting the correct insert. *UAS-LT3-Dam::Pcl* was as previously described in *Vaziri et al., 2020*. The *UAS-LT3-Dam::OGT*, *UAS-LT3-Dam::*Pcl, and *UAS-LT3-Dam* line were crossed to

the *Gr5a-GAL4; tubulin-GAL80^{ts}*. All animals were raised and maintained at 20°C. Expression of *Dam::OGT/Pcl* and *Dam* was induced at 28°C for 18 hr. For all experiments, 300 heads of male and female flies were collected per replicate on dry ice by sieving. DNA was extracted following kit instructions (Invitrogen). To identify methylated regions, purified DNA was digested by Dpn I, followed by PCR purification of digested sequences. TaDa adaptors were ligated by T4 DNA ligase (NEB). Adapter ligated DNA was PCR amplified and purified according to the protocol (*Marshall et al., 2016*). Purified DNA was digested with Dpn II, followed by sonication to yield fragments averaging 300 base pairs. TaDa adaptors were removed from sonicated DNA by digestion followed by PCR purification, and purified sonicated DNA was used for library preparation (*Vaziri et al., 2020*; *Marshall et al., 2016*).

   pUAST-Sxc.Forward         gatctgGCCGGCGCaATGCATGTTGAACAAACACGAATAAATATG, pUAST-Sxc.Reverse gttccttcacaaagatcctTTATACTGCTGAAATGTGGTCCGGAAG.

## Library preparation

Generation of RNA sequencing (RNA-seq) libraries was with the Ovation SoLo RNA-seq System for *Drosophila* (NUGEN, 0502-96). All reactions included integrated Heat-Labile Double-Strand Specific DNase treatment (ArcticZymes, catalog no. 70800-201). The DNA-sequencing libraries for TaDa were generated using the Takara ThruPLEX Kit (catalog no. 022818). For rat RNA-seq, libraries were prepared using the Nugen Ovation Model organism (Rat #0349-32) with 1/10th ERCC spike-in mix. These libraries were run on a NextSeq instrument using a HO 150 cycle cit (75 × 75 bp paired-end reads). All *Drosophila* libraries were sequenced on the Illumina NextSeq platform (High-output kit v2 75 cycles) at the University of Michigan Genomics Core facility.

## High-throughput RNA-seq analysis

Fastq files were assessed for quality using FastQC (*Andrews, Simon, and Others, 2010*). Reads with a quality score below 30 were discarded. Sequencing reads were aligned by STAR (*Dobin et al., 2013*) to dmel-all-chromosomes of the dm6 genome downloaded from Ensembl genomes. Counting was conducted by HTSeq (*Anders et al., 2015*). Gene counts were used to call differential RNA abundance by DESeq2 (*Love et al., 2014*). A pipeline was generated from *Wilinski et al., 2019*. To determine the efficiency and cell specificity of the TRAP, pairwise comparisons were made between the *Gr5a+*-specific fraction and the input. For comparisons between dietary conditions, DESeq2 was only applied to the *Gr5a+*-specific IP condition. SD7 and *Pcl^{c429}* datasets were analyzed from and described in *Vaziri et al., 2020*. A cutoff of *q* < 0.1 was used to call DEGs. To identify overlap between datasets GeneOverlap was used (*Shen and Sinai, 2013*).

## High-throughput TaDa and CATaDa analysis

Fastq files were assessed for quality using FastQC (*Andrews, Simon, and Others, 2010*). Reads with a quality score below 30 were discarded. The damidseq_pipeline was used to align, extend, and generate log2 ratio files (*Dam::OGT/Dam and Dam::Pcl/Dam*) in GATC resolution as described previously (*Marshall and Brand, 2015*). Reads were mapped by Bowtie2 (*Langmead and Salzberg, 2012*) to dmel-all-chromosomes of the dm6 genome downloaded from Ensembl genomes, followed by read extension to 300 bp (or to the closest GATC, whichever is first). Bam output is used to generate the ratio file in bedgraph format. Bedgraph files were converted to bigwig and visualized in the UCSC Genome Browser. Principal components analysis plots between biological replicates were computed by multibigwigSummary and plotCorrelation in deepTools (*Ramírez et al., 2016*). Peaks were identified from ratio files using find_peaks (FDR <0.01) (*Marshall and Brand, 2015*) and as in *Vaziri et al., 2020*. Overlapping intervals or nearby intervals (up to 50 bp) were merged into a single interval using mergeBed in BEDtools (*Quinlan and Hall, 2010*). Intervals common in at least two replicate peak files were identified by Multiple Intersect in BEDtools and used to generate the consensus peaks (*Quinlan and Hall, 2010*). For CATaDa experiments, all analyses were performed similarly to those of TaDa with the exception that *Dam* only profiles were not normalized as ratios but shown as normalized binding profiles generated by converting bam files to bigwig files normalized to 1× dm6 genome as reads per genome coverage (Sequencing depth is defined as the total number of mapped reads times the fragment length divided by the effective genome size). Binding intensity metaplots were made by computing a matrix for specified regions (*Ramírez et al., 2016*). To determine the proportion of genes that fit within the various chromatin domain subtypes, we first matched Dam::OGT/Dam targets to

coordinates identified by *Filion et al., 2010* and then determined their gene count in each chromatin subtype (observed) compared to the whole genome (expected). Peak annotations were conducted using the HOMER annotatePeaks tool (*Heinz et al., 2010*) with the dm6 reference genome. In TaDa analysis, genes were considered targets of the factor being investigated if a peak existed anywhere on their length.

## Pathway enrichment analysis

For all fly experiments, GO term enrichment analysis was performed using the iPAGE package (*Goodarzi et al., 2009*), using gene-GO term associations extracted from the Flybase dmel 6.08 2015_05 release. For all analyses, iPAGE was run in discrete mode. Independence filtering was deactivated for all discrete calculations. All other iPAGE settings default values. All shown GO terms pass the significance tests for overall information described in *Goodarzi et al., 2009*. For each term, bins that are outlined show especially strong contributions [p values such that a Benjamini–Hochberg FDR (*Benjamini and Hochberg, 1995*) calculated across that row yields $q < 0.05$].

## Analysis of cis-regulatory enrichments

For each *D. melanogaster* DNA-binding protein motif available from the CIS-BP database (*Weirauch et al., 2014*), we scanned the *D. melanogaster* genome (dmel 6.08 2015_05 release) using the FIMO-binding site discovery tool [cite:doi:10.1093/bioinformatics/btr614]. Hits for each motif were retained as potential binding sites and used to calculate overlaps with other features (e.g., OGT of Pcl sites), as noted. Permutation tests to assess significance were performed through repeated application of the bedtools shuffle [cite: https://doi.org/10.1093/bioinformatics/btq033] command to obtain 100 resamplings (*Supplementary file 1*) or 1000 resamplings (*Figure 4*) of the feature location of interest, requiring non-overlap of the randomly placed features. For the analysis in *Supplementary file 1*, we separately considered each potential motif for each transcription factor extracted from the CIS-BP database (separate motifs for the same factor are denoted by the gene name followed by a '_#' suffix, with # and integer). In the case of analysis of Sr-binding sites in *Figure 4*, we obtained a merged set of potential Sr-binding sites by filtering potential binding sites at a $q$ value threshold of 0.1 (acting separately for each motif) and combining all of the locations that were counted as a potential binding site for any of the Sr motifs available from CIS-BP. Enrichments of overlaps with OGT, Pcl, and PRE sites were calculated by comparing the actual observed count of overlapping features with the mean overlap observed across 1,000 random samplings of the Sr motif locations (preserving the chromosome on which each motif is located during shuffling). For comparison of Sr motif locations with TSSs, we first identified the (strandedness-aware) start location of all 'gene', 'mobile_genetic_element', or 'pseudogene' features from the dmel6 Genbank annotations and then categorized all of these locations as 'OGT/PRC2' or 'Not OGT/PRC2' based on whether or not the gene was associated with an OGT, Pcl, or PRE location (see *Supplementary file 1* for gene lists for each feature type). The density of Sr motif hits (as defined above) was then calculated as a function of position relative to the TSS.

## Electrophysiology

Extracellular recording on labellar sensilla was performed using the tip recording method (*Delventhal et al., 2014*). Ten- to thirteen-day-old flies were anesthetized by short ice exposure. The reference electrode containing the Beadle–Ephrussi Ringer solution was inserted through the thorax into the labellum to immobilize the proboscis. The neuronal firing in L-type sensilla was recorded with a recording electrode (10–20 µm diameter) containing 25 mM sucrose dissolved in 30 mM tricholine citrate as an electrolyte. The recording electrode was connected to TastePROBE (Syntech), and electrical signals were obtained using the IDAC acquisition controller (Syntech). The signals were amplified (10×), band-pass-filtered (100–3000 Hz), and sampled at 12 kHz. Neuronal firing rates were analyzed by counting the number of spikes for a 500-ms period starting from 200 ms after contact using the Autospike 3.9 software. Experimenters were blinded in the initial characterization of the phenotypes and experiments were independently performed at least three times.

## Data analysis and statistics

Statistical tests, sample size, and *p* or *q* values are listed in each figure legend. One- or two-way repeated measure analysis of variance with post hoc tests were used for all PER experiments. All

behavioral data were tested for normality, and the appropriate statistical tests were applied if data were not normally distributed. For the RNA-seq expression datasets, we coupled our standard differential expression with a test for whether each gene could be flagged as 'significantly not different' – that is, a gene for which we can confidently state that no substantial change in expression occurred (rather than just a lack of evidence for change, as would be inferred from a large p-value on the differential expression test). Defining a region of practical equivalence as a change of no more than 1.5-fold in either direction, we tested the null hypothesis of a change larger than 1.5-fold using the gene-wise estimates of the SE in log2fold change (reported by Deseq2) and the assumption that the actual l2fcs are normally distributed. Rejection of the null hypothesis is evidence that the gene's expression is not changed substantially between the conditions of interest. Python code for the practical equivalence test is in *Source code 1*. All data in the figures are shown as means ± standard error of the mean, ****p < 0.0001, ***p < 0.001, **p < 0.01, and *p < 0.05, unless otherwise indicated.

## Data and material availability statement

All high-throughput data are available at the GEO repository: GSE188757 and GSE146245. *LT3-Dam::OGT* and *Pcl* flies are available upon request; all other fly lines are available in the BDSC database as shown in *Supplementary file 1*.

## Acknowledgements

We thank the University of Indiana at Bloomington, the VDRC, the FLYORF stock collections, and all the investigators who shared fly lines with us. Julia Kuhn designed some of the graphics for the manuscript. We are grateful to Dr. Morteza Khabiri for assistance in the calculation of potential TF-binding sites. This work was funded by NIH R00 DK-97141, NIH 1DP2DK-113750, NIH R01DK130875, the Klingenstein-Simons Fellowship in the Neurosciences, the Rita Allen Foundation, and NSF CAREER 1941822 (all to MD), the Rackham Predoctoral Fellowship (to AV), NIH T32 DA007268 (DW), NIH K99 DK128539 (DW), NIH P30 DK089503 (MD and DW), and NIH R35GM128637 (to PLF).

## Additional information

### Funding

| Funder | Grant reference number | Author |
| --- | --- | --- |
| National Institute of Diabetes and Digestive and Kidney Diseases | R00 DK-97141 | Monica Dus |
| National Institute of Diabetes and Digestive and Kidney Diseases | DP2DK-113750 | Monica Dus |
| National Institute of Diabetes and Digestive and Kidney Diseases | R01DK130875 | Monica Dus |
| National Institute of Diabetes and Digestive and Kidney Diseases | T32 DA007268 | Daniel Wilinski |
| National Institute of Diabetes and Digestive and Kidney Diseases | DK128539 | Daniel Wilinski |
| National Science Foundation | CAREER 1941822 | Monica Dus |
| National Institute of Diabetes and Digestive and Kidney Diseases | P30 DK089503 | Monica Dus |
| National Institute of General Medical Sciences | R35GM128637 | Lydia Freddolino |

| Funder | Grant reference number | Author |
|---|---|---|
| Esther A. and Joseph Klingenstein Fund | Fellowship in the Neurosciences | Monica Dus |
| Rita Allen Foundation | Scholar | Monica Dus |
| Klingenstein-Simons Fellowship | | Monica Dus |
| Rackham Predoctoral Fellowship | | Anoumid Vaziri |

The funders had no role in study design, data collection, and interpretation, or the decision to submit the work for publication.

### Author contributions
Hayeon Sung, Conceptualization, Investigation, Methodology, Writing – review and editing; Anoumid Vaziri, Conceptualization, Resources, Formal analysis, Investigation, Visualization, Methodology, Writing – original draft; Daniel Wilinski, Resources, Formal analysis, Writing – review and editing; Riley KR Woerner, Formal analysis, Investigation, Writing – review and editing; Lydia Freddolino, Data curation, Software, Investigation, Visualization; Monica Dus, Conceptualization, Formal analysis, Supervision, Funding acquisition, Investigation, Visualization, Writing – original draft, Project administration, Writing – review and editing

### Author ORCIDs
Hayeon Sung ⓘ http://orcid.org/0000-0002-9015-8877
Riley KR Woerner ⓘ http://orcid.org/0009-0000-0702-138X
Lydia Freddolino ⓘ https://orcid.org/0000-0002-5821-4226
Monica Dus ⓘ http://orcid.org/0000-0003-1465-9028

### Decision letter and Author response
Decision letter https://doi.org/10.7554/eLife.83979.sa1
Author response https://doi.org/10.7554/eLife.83979.sa2

## Additional files

### Supplementary files
Supplementary file 1. Excel file with the analyses of the DAM and TRAP experiments.

MDAR checklist

Source code 1. Python script to run the practical equivalence test (see methods).

### Data availability
Sequencing data have been deposited in GEO under accession codes: GSE188757 and GSE146245. Analyzed omics data are included in *Supplementary file 1*.

The following dataset was generated:

| Author(s) | Year | Dataset title | Dataset URL | Database and Identifier |
|---|---|---|---|---|
| Sung and Vaziri | 2022 | A nutrient information pathway links diet to taste | http://www.ncbi.nlm.nih.gov/geo/query/acc.cgi?acc=GSE188757 | NCBI Gene Expression Omnibus, GSE188757 |

The following previously published dataset was used:

| Author(s) | Year | Dataset title | Dataset URL | Database and Identifier |
|---|---|---|---|---|
| Vaziri A, Khabiri M, Genaw BT, May CE | 2020 | Persistent Epigenetic Reprogramming of Sweet Taste by Diet | https://www.ncbi.nlm.nih.gov/geo/query/acc.cgi?acc=GSE146245 | NCBI Gene Expression Omnibus, GSE146245 |

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
