## [Editor Report]

Using *Drosophila* gustatory neurons as a model system, the authors provide important mechanistic insight into how nutrigenomic signaling encodes nutritional information into cellular changes. The authors expand previous work by showing that OGT is associated with neural chromatin at introns and transcriptional start sites and that diet-induced changes in chromatin accessibility were amplified at loci with the presence of both OGT and PRC2.1. The work also identifies Mitogen Activated Kinase as a critical mediator in this pathway. This is an elegant group of experiments revealing mechanisms for how nutrigenomic signaling triggers cellular responses to nutrients.

---

## [Decision Letter]

**Decision letter after peer review:**

Thank you for submitting your article "Nutrigenomic Regulation of Sensory Plasticity" for consideration by *eLife*. Your article has been reviewed by 3 peer reviewers, and the evaluation has been overseen by a Reviewing Editor and K VijayRaghavan as the Senior Editor. The following individual involved in review of your submission has agreed to reveal their identity: Allan-Hermann Pool (Reviewer #3).

Essential revisions:

There was relative consensus amongst the reviewers about the interest and potential importance of this study. In addition to the relatively straightforward controls and textual revisions detailed below, there were two major areas in which additional experiments are needed to strengthen and extend the impact of this study.

1) Perform additional genetic experiments (detailed below) to confirm and underscore the specificity of the pharmacological experiments. This might include a demonstration that the drug phenocopies the OGT enzymatic loss-of-function and related experiments.

2) Presenting analyses in non-sweet sensing neurons to strengthen the impact and specificity of the manipulations.

*Reviewer #1 (Recommendations for the authors):*

As the study currently stands, the proposed model is speculative, and a significantly revised approach and design would be needed to support the study's conclusions.

1) The genetic interactions of OGT with other genes (PRC, Plc, rolled) rely only on the treatment of the flies with a systemic pharmacological OGT activity inhibitor OSMI. But, it is not possible to draw strong and nuanced conclusions from the datasets that rely heavily of a systemic drug-based manipulation instead of precise genetic alleles. Ideally the authors would want to include the same DAM-ID experiments performed with a OGT enzymatic mutant versus a OGT-WT.

The authors are encouraged to ask what happens to PRC occupancy in CD and SD in tissue-specific KD of OGT using RNAi. If the RNAi reagent does not provide the authors a complete loss of function, they should consider using sgRNA-based KD approaches or generate a HA or GFP OGT knock-in line and then utilize degron based tissue-specific KD of OGT.

2) For the OSMI experiments, would the authors present evidence that the drug is specific to OGT and has no off-target effects? Alternatively, authors might want to consider presenting results with OGT KD or OGT enzyme specific mutants to correlate OSMI results with. These orthogonal and corroborative datasets would be required to bolster the conclusions.

3) This study is measuring the chromatin availability and determined that there was higher accessibility under CD conditions. Since OGT also has critical cytosolic roles, it would be important to determine if this is due to differences in nuclear: cytoplasmic localization of OGT or due to changes in OGT activity. To this end, I would recommend a western blot of Nuclear v. cytoplasmic fractions of OGT for flies on CD vs SD or imaging based approaches would be needed.

4) In Figure 2A, the authors describe a high correlation in chromatin binding sites between Pcl and OGT, concluding that the overlapping OGTXPRC2.1 regions are a critical subset for sweet tasting response. However, these overlapping regions are a rather small subset of sites, with the largest subset actually belonging the "yellow" H3K4/H3K36 euchromatin subset. It is unclear from this analysis why the authors chose to pursue the blue chromatin instead. It would be important to rank regulatory regions bound by DAM:OGT alone and DAM:Pcl alone to establish the primary regions of binding for both OGT and Pcl irrespective of each other. These regions can then be compared with the known sweet tasting genes to rank the relevant binding regions. Should there be other regions relevant to sweet taste that are not overlapping between OGT and Pcl that should be reported. This will also give readers a more comprehensive understanding about the role of Sr motifs and if this is a small but significant subset of genes regulated by OGT or comprise the majority of cis-regulatory regions controlled by OGT.

5) The authors use PCL as a proxy for PRC2.1 activity as it is the recruiter for PRC2.1. However, why wasn't DAM::PRC2.1 performed to directly compare to OGT? This data should be provided to show the overlap of PRC2.1 to Pcl or it should be explained in the results.

6) The authors findings are performed using flies overexpressing Pcl or OGT in Gr5a expressing cells for the TADA, PER experiments, and electrophysiology analysis. Because this analysis is based on an overexpression of the gene of interest, I believe it is important to repeat these experiments in flies overexpressing these genes in neurons unrelated to the sweet taste response- such as in the neighboring bitter taste neurons. This is especially a critical set of experiments to do, as the primary thesis of this study is about how each cell alters its nutrigenomic signaling in different cell types, but the analysis is primarily focused on DAM-ID datasets from overexpressed OGT in Gr5a. The study would be relevant if the authors did comparative analysis of OGT-PRC interactions in neurons that have different sensitivities or taste modalities to sugar.

7) The Proboscis extension response needs to be further elaborated on. The data plotted seems to be normalized but this is not explained in the paper. What is actually being measure? What is the N? If the data is normalized, then the authors should present raw data graphed in a supplemental figure at least.

8) Figure 4B seems misleading in the way it is plotted. Authors claim a bias of SR genes at the TSS in the PRC2.1 and OGT overlapping regions. However, the y axes are not to the same scale. In fact, the not OGT/PRC2 regions have much higher number of genes with SR motifs at every single position plotted on 4B. Specifically the strong statements findings in the results paragraph on lines 262-283 are not sufficiently supported and the authors are encouraged to revise this or move it to discussion.

9) Many of the experiments involved moving flies to OSMI or Trametinib diets. A caveat to this method is that this treatment will influence OGT/ERK in the fly globally. Because both OGT/ERK impact a number of critical biological processes, this makes the results difficult to interpret as impacts of taste sensing or due to the flies being generally unhealthy. It would be helpful to see fly survival on control, OSMI supplemented, and Trametinib diets to determine how detrimental these diet conditions are in general to the fly.

10) In general, the figure legends and methods are lacking in detail. For instance, in Figure 1F, are the GO terms for control diet comprised from all peaks listed or just the peaks unique to CD? Same question for the SD GO terms. Abbreviations are not elaborated in the legend or methods (ex: Figure 2B TPM) or lacking axis labels (ex: Figure 2D), etc. I suggest a thorough review of the methods and legend to add details necessary to interpret the results and make it possible to replicate experimental designs.

*Reviewer #2 (Recommendations for the authors):*

1. There are too many instances of non-technical phrases that are unclear/nonspecific. For example, "…OGT moonlights on the chromatin ….", or "… OGT decorates nutrient-sensitive …". These phrases should be replaced with more precise words, unless these phrases are common in the field and I am just unaware.

2. The Discussion section is overall too long and delves too much into the previous literature. It should be condensed and there should be more focus on the translational implications of the results.

*Reviewer #3 (Recommendations for the authors):*

– Almost all quantitative data in figures lack definition of variance measures (e.g. Panel 3C and E). Are data represented as mean +/I standard deviation, s.e.m.?

– Regarding setting up the problem – claiming "how nutritional information is transmuted to neural activity and behavioral changes remains elusive" is inaccurate – we know the receptors, increasingly many cells in the food intake regulatory circuit from sensory input to motor output in *Drosophila*.

– Methods refer to a python code on Github but no github repository is specified.

– Please also make the code for analyzing the DaMID and CaTaDa available online.

---

## [Author Response]

Essential revisions:There was relative consensus amongst the reviewers about the interest and potential importance of this study. In addition to the relatively straightforward controls and textual revisions detailed below, there were two major areas in which additional experiments are needed to strengthen and extend the impact of this study.1) Perform additional genetic experiments (detailed below) to confirm and underscore the specificity of the pharmacological experiments. This might include a demonstration that the drug phenocopies the OGT enzymatic loss-of-function and related experiments.

We have carried out several new genetic experiments to address the reviewers' concerns. In the revised manuscript, these experiments are shown in a new Figure 3 and revised Figure 4, and Figure 4-Supplement 1, Figure 5-Supplement 1, and Figure 6-Supplement 1; below, we outlined the experiments and explained the findings for the reviewers’ below before addressing the individual comments.

1A. OGT catalytic activity and OSMI-1

There are several published OGT mutant alleles, but we were only able to obtain two from the lab that generated them; we were told that all the others, including the UAS-versions which would have been useful for our experiments, were lost due to the mishandling of the fly stocks. We requested these stocks in early 2020 but were unable to get them for nearly a year due to the COVID lab restrictions in the UK; this is why we originally carried out the experiments with OSMI-1 once our own lab reopened at 20% capacity in July 2020-May 2021 since we had previously validated, used, and published with this reagent (May et al. 2020). In brief, OSMI-1 is a specific inhibitor of OGT developed by Harvard Medical School and the NIH and published in 2015 (Ortiz-Meoz et al. 2015); its inhibition is insensitive to UDP-GlcNAc concentration, and it doesn’t affect cell-surface glycosylation.

In data now added to Figure 3C, we compared *OGT^K872M^*, a full catalytically dead mutant, to the protein null *OGT^1 –^* please note that both of these are homozygous lethal, so here they were tested in heterozygous combination with *w1118cs* (*wcs*) control flies (Mariappa et al. 2015). As you can see in Figure 3, the heterozygous catalytically dead mutant, *OGT^K872M/+^* (100% of OGT protein but only 50% catalytic activity, see diagram on the right), phenocopies the effects of protein null *OGT^1^* allele (50% of protein with catalytic activity) in preventing diet-induced taste plasticity on a sugar diet (SD, squares). No effects were observed on a control diet (CD, circles). This provides genetic evidence for OGT activity being essential for taste changes in response to a sugar diet. Please note that as the diagram in Figure 3A now shows, proboscis extension response (PER) is measured from 1 (full extension) to 0 (no extension). Thus, the PER data shown in the manuscript is the raw data (mean +/-SEM) obtained from observing the flies during sucrose stimulation; it is not normalized (a comment of Reviewer 1). Importantly, the phenotype of *OGT^K872M^*^/+^ flies (wild-type taste on sugar diet) is similar to that of *OGT RNAi* targeted specifically to the Gr5a+ neurons (now Figure 3B). It is also identical to that of OSMI-1 treatment (now Figure 3F).

To further validate the role of OGT catalytic activity, we knocked down the gene for the antagonistic enzyme to OGT, *O-GlcNAcase* (*OGA*), which removes the GlcNAc moiety. If OGT catalytic activity were important, we would expect that interfering with removing the GlcNAc mark would mimic the effects of a sugar diet (and OGT overexpression) on taste. Figure 3 D shows that the knockdown of *OGA* in the Gr5a+ neurons resulted in a marked decrease in sucrose taste sensation in control-diet flies (Figure 3D). Thus, *OGA* knockdown recapitulates the effects of OGT overexpression and sugar-diet on sucrose taste, further validating the role of OGT activity in taste plasticity.

Finally, to link OSMI-1 to OGT function in our system, we tested the effects of OSMI-1 treatment in flies that overexpress *OGT* in the Gr5a+ neurons. As a reminder, expression of *OGT* in the Gr5a+ neurons results in taste depression on a control diet, recapitulating the effects of a sugar diet. Treatment with OSMI-1, however, inhibited the effects of OGT overexpression on sweet taste, as shown in Figure 3E. These results link the OGT overexpression phenotype to its catalytic activity. As reviewer 1 asked, we tested the effects of OSMI-1 and Tretmetinib on survival and found no effect (Figure 4-Supplement 1A).

Together, the experiments in the new Figure 3 provide solid evidence for the role of OGT catalytic activity in taste plasticity and support the design of the molecular experiments in Figure 4.

1B. Epistasis experiments

Reviewers 2 and 3 commented on the “elegant group of experiments” and “the thorough exploration of epistatic relationships between different players in the system.” These epistatic relationships would be very hard – and perhaps impossible– to parse out without the combination of genetic and pharmacological tools we used here. Since Reviewer 1 expressed concern about OGT, Pcl, and PRC2 interactions, we carried out additional genetic and pharmacological epistasis experiments, which we added to Figure 4 in the revised manuscript.

Figure 4C shows that the taste phenotype caused by *OGT* overexpression in the Gr5a+ neurons (green) is dependent on the presence of a functioning *Pcl* (pink and purple). And consistent with the function of Pcl as a PRC2.1 recruiter, Figure 4D shows that the ability of OGT to induce taste plasticity depends on the catalytically active PRC2 complex. These data provide genetic evidence for the data shown in the original version of the manuscript (now Figure 4B), where we rescued the effects of *Pcl* overexpression on taste with OSMI; they also suggest that PRC2 H3K27 methylation is likely the effector arm of this nutrigenomic signaling. A note in response to Reviewer 1: we manipulated Pcl instead of Ez because this manipulation is more specific. The PRC2 complex comes in many different “flavors,” all of which contain Ez as the H3K27m. Each of the PRC2 variants is characterized by a different recruiter, which targets the complex to specific regions of chromatin (i.e., Pcl has a Tudor domain which is thought to target PRC2.1, the PCl^-^containing PRC2, to actively transcribed chromatin). Thus, by manipulating Pcl, we can target a specific type of PRC2.1; in (Vaziri et al. 2020) we demonstrated that the other PRC2 variants are not involved in taste plasticity; in that manuscript, we also show that *Pcl, Ez, Suz12,* and *Esc* mutants phenocopy one another, and that the effects of Pcl on sweet taste are dependent on the catalytic activity of Ez.

Together these new data in Figure 4 place OGT and PRC2.1 in the same genetic pathway, and argue for a strong interaction between OGT, Pcl, and PRC2; these results also support those of the DAM-ID and TRAP molecular experiments, which show synergistic effects between these two players.

Additional experiments

We added data showing that Gr5a+-specific knockdown of the kinase *rl* (we previously only had mutant data) has marked effects on sucrose taste sensitivity (Figure 6-Supplement 1)

We also added a functional characterization as measured by proboscis extension response of some of the sugar diet DEGs that are dependent on OGT/Pcl as suggested by Reviewer 3 (Figure 4).

These genetic experiments make the conclusions of the manuscript stronger – we thus hope the reviewers find their concerns addressed.

2) Presenting analyses in non-sweet sensing neurons to strengthen the impact and specificity of the manipulations.

This study is a cell-specific investigation of nutrigenomic mechanisms in the Gr5a+ sweet-taste neurons; our intention was not to examine mechanisms across different cell types. However, we can understand the reviewers’ comments about this after rereading the abstract and introduction. As such, we have rewritten part of the manuscript to better introduce the rationale behind the study as the integration of metabolic signaling and cellular contexts. We hope this is now an improved framing for the study rationale.

About analyzing the effects of diet on other cells, no doubt this is an interesting question. However, this also signifies embarking on a completely separate project that would take, optimistically speaking, at least one year to complete and require a budget of ~ $130,000 (see breakdown). Thus, this suggestion doesn’t seem in line with the peer review and editorial philosophy of *eLife*. Carrying out this new project would result in an additional 6-7 figures but would not fundamentally change the conclusion of the current work; in fact, it may even take away from the targeted integration of molecular biology and neuroscience we have tried to achieve.

Moreover, talking about the practicality of this proposal, we do not have such an unallocated budget, and so this new project would require us first to generate preliminary data on the bitter neurons, then write a grant proposal to fund it; as you can appreciate, this would take longer than a year, especially since we do not even know if the bitter gustatory neurons are affected by a high-sugar diet. Beyond this, looking at the bitter neurons would do little to prove specificity. If we found no effects of this pathway on the activity of the bitter neurons, it wouldn’t establish that the changes in the sweet taste neurons are specific. In fact, the same pathway could be acting in some of the other thousands of fly circuits that were not investigated (Black swan effect). If we did find that OGT/PRC2.1/Sr play a role in the bitter neurons, it would also do little to disprove specificity since their targets would likely be different because the sets of genes expressed in these two sensory neurons are different. By analogy, the protein sensor mTOR is expressed and active in every cell, where it modulates some of the same targets (i.e., S6K); however, the effects of the pathway may be different due to the distinct metabolic and genetic idiosyncrasies of cells, as well as cellular compartments. This lack of specificity doesn’t mean that mTOR is not important or unspecific. Finally, we would like to note that we have tested the effects of manipulating OGT levels in other neurons (dopamine and Mushroom Body Output Neurons) without effects on behavior or neural responses (May et al. 2020; Pardo-Garcia et al. 2022); based on these, OGT doesn’t seem to affect neurons indiscriminately.

Budget ~ $129,000

Salary and fringe benefits for PD for 10 calendar months: $75,000 (2 months behavior experiments, 2 months training for molecular biology experiments and troubleshooting in new neurons, 4 months growing flies and conducting experiments, 2 months data analysis and visualization)

DAM ID: $16,0000, Pcl:dam and OGT:dam in CD and SD, with and without OSMI x 4 biological replicates per condition = 32 samples @ $500 per sample (UM Genomics core)

TRAP: $16,0000 Pcl mutant and OSMI in CD and SD x 4 biological replicates per condition + sequencing input = 32 samples @ $500 per sample (UM Genomics core)

Animals' food and maintenance: $ 5,000, $500 per person/10 months

Reagents: = $17,000, including sequencing kit (32 reactions = $6,000) x 2 = $12,000, and other reagents such as drugs and plastic

Note that this PD would have to be hired and retrained. The first author of the manuscript who carried out the molecular experiments graduated in Dec 2021 but failed to pass on the technical knowledge due to COVID restrictions at the UM: we were completely shut down until July 2020, and at 20% capacity from March 2020 to July 2021 (people also couldn’t work together to show techniques), and no new people joined the lab in 2020-2022 (most of the 2021 grad student class deferred to 2022 and no PD candidates apply).

Reviewer #1 (Recommendations for the authors):As the study currently stands, the proposed model is speculative, and a significantly revised approach and design would be needed to support the study's conclusions.1) The genetic interactions of OGT with other genes (PRC, Plc, rolled) rely only on the treatment of the flies with a systemic pharmacological OGT activity inhibitor OSMI. But, it is not possible to draw strong and nuanced conclusions from the datasets that rely heavily of a systemic drug-based manipulation instead of precise genetic alleles. Ideally the authors would want to include the same DAM-ID experiments performed with a OGT enzymatic mutant versus a OGT-WT.The authors are encouraged to ask what happens to PRC occupancy in CD and SD in tissue-specific KD of OGT using RNAi. If the RNAi reagent does not provide the authors a complete loss of function, they should consider using sgRNA-based KD approaches or generate a HA or GFP OGT knock-in line and then utilize degron based tissue-specific KD of OGT.2) For the OSMI experiments, would the authors present evidence that the drug is specific to OGT and has no off-target effects? Alternatively, authors might want to consider presenting results with OGT KD or OGT enzyme specific mutants to correlate OSMI results with. These orthogonal and corroborative datasets would be required to bolster the conclusions.

Response to 1 and 2 (as point 1 in the author’s recommendations):

In data now added to Figure 3C and added here for the reviewer’s convenience, we show that heterozygous catalytically dead mutants, *OGT^K872M^/+* ((Mariappa et al. 2015) 100% of OGT but only 50% cat activity, see diagram on the right), phenocopies the effects of protein null *OGT^1^* allele (50% of protein with catalytic activity) in preventing diet-induced taste plasticity on a sugar diet (SD). No effects were observed on a control diet (CD). This shows genetic evidence for OGT activity being essential for taste changes in response to a SD. Please note that as the diagram in Figure 3A now shows, proboscis extension response (PER) is measured from 1 (full extension) to 0 (no extension). Thus, the PER data shown in the manuscript is the raw data (mean +/-SEM) obtained from observing the flies during sucrose stimulation; it is not normalized (see below for full response to this comment). Importantly, the phenotype of *OGT^K872M^/+* flies (wild-type taste on sugar diet) is similar to that of OGT RNAi targeted specifically to the Gr5a+ neurons (now Figure 3B). It is also identical to that of the OSMI-1 treatment (now Figure 3E). OSMI-1 is a specific inhibitor of OGT developed by Harvard Medical School and the NIH and published in 2015 (Ortiz-Meoz et al. 2015); its inhibition is insensitive to UDP-GlcNAc concentration, and it doesn’t affect cell-surface glycosylation. We had previously used OGT in (May et al. 2020).

If OGT catalytic activity were important, we would expect that interfering with the removal of GlcNAcylation would mimic the effects of a sugar diet and OGT overexpression on taste. We tested this possibility by knocking down the antagonistic enzyme to OGT, O-GlcNAcase (OGA), which removes the GlcNAc moiety only in the Gr5a+ neurons. This resulted in a marked decrease in sucrose taste sensation in control-diet flies (Figure 3D), mimicking the effects of *OGT* overexpression and sugar diet and further validating the role of OGT activity.

To further validate these findings, we tested the effects of OSMI-1 treatment in flies that overexpress OGT in the Gr5a+ neurons. As a reminder, expression of OGT in the Gr5a+ neurons results in taste depression on a control diet, mimicking the effects of a sugar diet. Treatment with OSMI-1, however, inhibited the effects of OGT overexpression on taste, as shown in Figure 3 E. These results link the OGT overexpression phenotype to its catalytic activity.

Together, the experiments in the new Figure 3 provide further evidence for the role of OGT catalytic activity in taste plasticity.

We disagree with the comment that we rely solely on pharmacological manipulations; in fact, we use cell-specific – pharmacological and dietary interventions. Indeed, Reviewers 2 and 3 praised our approaches by saying: “This is an elegant group of experiments revealing mechanisms for how nutrigenomic signaling triggers cellular responses to nutrients” and “Strengths: the thorough exploration of epistatic relationships between different players in the system.” All of our experiments combine genetic manipulations in combination with dietary and/or pharmacological treatments to show that molecular, neural, and behavioral taste phenotypes arise only in specific contexts, so no single phenotype occurs due to nonspecific manipulations. Without this approach, most of these epistatic relationships would be largely inaccessible in this system. We have also used a combination of both genetic and pharmacological tools to implicate not only genes but also their function (i.e., enzymatic activity) to nutrient-specific effects. Third, we established causality and relationship by inducing and rescuing the molecular, behavioral, and electrophysiological phenotypes. Thus, our model is based on a combination of direct and indirect data (genetic manipulations are by nature inferential) obtained from a controlled and careful set of experiments.

We could spend the next 6+ months trying to generate cell-specific alleles of OGT, which could bring their own unspecific effect, such as development and compensatory effects. However, this would only marginally add to the story now that we have added additional characterizations. In particular, we carried out additional genetic and pharmacological epistasis experiments, which we added to Figure 4. Here, we show that the taste phenotype caused by OGT overexpression in the Gr5a+ neurons entirely depends on the presence of *Pcl*. This provides genetic evidence for the data in the original manuscript version (now Figure 4B), where we rescued the effects of Pcl overexpression with OSMI-1.

We manipulated *Pcl* instead of *Ez* because this manipulation is more specific. The PRC2 complex comes in many different “flavors,” all of which contain Ez. Each of the PRC2 variants is characterized by a different recruiter, which targets the complex to specific regions of chromatin (i.e., Pcl has a Tudor domain which is thought to target PRC2.1, the PCl^-^containing PRC2, to actively transcribed chromatin). Thus, by manipulating *Pcl*, we can target a specific type of PRC2.1, and as we showed in, (Vaziri et al. 2020), the other PRC2 variants do not play a role in taste plasticity; in that manuscript, we also show that *Pcl, Ez, Suz12,* and *Esc* mutants phenocopy one another, and that the effects of Pcl on taste are indeed dependent on the catalytic activity of Ez. Here we took a similar approach and showed that *OGT* expression requires not only *Pcl*, but a catalytically active PRC2 to mediate its effect on taste plasticity (Figure 4D).

Together the new Figure 4 experiments place OGT and PRC2.1 in the same genetic pathway, indicating a strong interaction between OGT, Pcl, and PRC2, and support the results of the DAM-ID and TRAP molecular experiments, which show synergistic effects between these players.

Finally, we added data showing that Gr5a+-specific knockdown of the kinase *ERK* (we previously only had mutant data) has marked effects on sucrose taste sensitivity (now in Figure 6-Supplement 1C).

The addition of these genetic experiments makes the manuscript stronger and hope the reviewer finds their concerns addressed.

3) This study is measuring the chromatin availability and determined that there was higher accessibility under CD conditions. Since OGT also has critical cytosolic roles, it would be important to determine if this is due to differences in nuclear: cytoplasmic localization of OGT or due to changes in OGT activity. To this end, I would recommend a western blot of Nuclear v. cytoplasmic fractions of OGT for flies on CD vs SD or imaging based approaches would be needed.

We agree with the reviewer that this is an important possibility. We tried available OGT antibodies for Western blot and immunofluorescence (to retain cell specificity), but they were not functional. In 2019 we attempted to generate rabbit polyclonal antibodies against OGT based on the two peptides below, as recommended by protein chemists:

DmOGT(570-592): KPYNFLKKLPTKGRLRIGYLSSD

DmOGT(1045-1059): Cys-EKYENGELPDHISAV

We injected 4 different rabbits, which gave rise to two different polyclonals per peptide. To our disappointment, none of the found antibodies showed a specific signal in Western blot and immunofluorescence experiments. We originally mentioned OGT compartmentalization in the discussion, but we agree that this point needs more stress, so we added it to the “limitation” portion of the discussion.

4) In Figure 2A, the authors describe a high correlation in chromatin binding sites between Pcl and OGT, concluding that the overlapping OGTXPRC2.1 regions are a critical subset for sweet tasting response. However, these overlapping regions are a rather small subset of sites, with the largest subset actually belonging the "yellow" H3K4/H3K36 euchromatin subset. It is unclear from this analysis why the authors chose to pursue the blue chromatin instead. It would be important to rank regulatory regions bound by DAM:OGT alone and DAM:Pcl alone to establish the primary regions of binding for both OGT and Pcl irrespective of each other. These regions can then be compared with the known sweet tasting genes to rank the relevant binding regions. Should there be other regions relevant to sweet taste that are not overlapping between OGT and Pcl that should be reported. This will also give readers a more comprehensive understanding about the role of Sr motifs and if this is a small but significant subset of genes regulated by OGT or comprise the majority of cis-regulatory regions controlled by OGT.

The OGT:dam data show that this enzyme is found at about an equal number of genes in yellow and blue chromatin (Figure 1B-D); only genes in blue chromatin are enriched for neural functions- see new text based on the suggestion to carry out more analyses on these two types of chromatin. The overlap between Pcl and OGT only occurs in a small proportion (162) of these OGT peaks, as shown in Figure 2. As we wrote in the text, the OGT/Pcl peaks are enriched in blue chromatin (permutation test); this is somewhat expected in a way since Pcl occupies blue chromatin, as shown in our previous analyses (Vaziri et al. 2020), and also demonstrated by the lower expression of the OGT x Pcl genes compared to OGT-alone (Figure 2B). Intriguingly, in (Vaziri et al. 2020) we observed that PCl^-^associated genes have a lower expression than non-PCl^-^bound genes but also show considerable variation in expression among them. The comparison in Figure 4B with Pcl vs. OGT/Pcl shows that the PCl^-^only genes have lower expression.

We added the enrichment for cis-regulatory regions among OGT-bound, PCl^-^bound, and OGT/Pcl and OGT/PREs regulatory regions in the supplementary excel file; the breakdown for Sr is shown in Figure 4A (now 5A). Very few genes are known as “sweet taste genes” outside of the sweet taste receptors, and the occupancy and expression of these sweet taste receptor genes is unchanged by diet. However, our previous molecular characterization of the sweet sensing neurons did identify two transcription factors, Ptx1, and cad, which we show here are occupied by OGT and Pcl, as mentioned in the manuscript. We took the reviewer’s suggestion and analyzed the GO terms and accessibility for OGT occupancy among yellow and blue chromatin (Figure 1-Supplement 1). We found that only blue genes showed enrichment in GO terms, which are a subset of those found in the OGT/Pcl genes. We discuss the results of the analysis in the text.

5) The authors use PCL as a proxy for PRC2.1 activity as it is the recruiter for PRC2.1. However, why wasn't DAM::PRC2.1 performed to directly compare to OGT? This data should be provided to show the overlap of PRC2.1 to Pcl or it should be explained in the results.

We manipulated and measured *Pcl* binding instead of *Ez* because this manipulation is more specific. The PRC2 complex comes in many different “flavors,” all of which contain Ez. Each of the PRC2 variants is characterized by a different recruiter, which targets the complex to specific regions of chromatin (i.e., Pcl has a Tudor domain which is thought to target PRC2.1, the PCl^-^containing PRC2, to actively transcribed chromatin). Thus, by manipulating and targeting Pcl, we can isolate a specific type of PRC2.1, and as we showed in (Vaziri et al. 2020), the other PRC2 variants do not play a role in taste plasticity; in that manuscript, we also show that *Pcl, Ez, Suz12,* and *Esc* mutants phenocopy one another, and that the effects of Pcl on taste are indeed dependent on the catalytic activity of Ez. In the revised version of Figure 4, we now show that *OGT* expression requires *Pcl* and a catalytically active PRC2 to mediate its effect on taste plasticity (4D).

6) The authors findings are performed using flies overexpressing Pcl or OGT in Gr5a expressing cells for the TADA, PER experiments, and electrophysiology analysis. Because this analysis is based on an overexpression of the gene of interest, I believe it is important to repeat these experiments in flies overexpressing these genes in neurons unrelated to the sweet taste response- such as in the neighboring bitter taste neurons. This is especially a critical set of experiments to do, as the primary thesis of this study is about how each cell alters its nutrigenomic signaling in different cell types, but the analysis is primarily focused on DAM-ID datasets from overexpressed OGT in Gr5a. The study would be relevant if the authors did comparative analysis of OGT-PRC interactions in neurons that have different sensitivities or taste modalities to sugar.

The expression of Pcl and OGT for the molecular experiment only occurs for 18 hours. We do not solely use overexpression, but also RNAi, loss of function alleles, and genetic manipulations compared to overexpression. Importantly, we show that our manipulations only have an effect in combination with an environmental variable (diet, i.e., OGT RNAi, OGT mutants, and Pcl mutants).

(As in the authors’ recommendation response): This study is a cell-specific investigation of nutrigenomic mechanisms in the Gr5a+ sweet taste neurons; it was not our intention to examine mechanisms across different cell types. However, we can understand the reviewers’ comments about this after rereading the abstract and introduction. As such, we have rewritten part of the manuscript to better introduce the rationale behind the study as the integration of metabolic signaling and cellular contexts. We hope this is now an improved framing for the study rationale.

About analyzing the effects of diet on other cells, no doubt this is an interesting question. However, this also signifies embarking on a completely separate project that would take, optimistically speaking, at least one year to complete and require a budget of ~ $130,000 (see breakdown). Thus, this suggestion doesn’t seem in line with the peer review and editorial philosophy of *eLife*. Carrying out this new project would result in an additional 6-7 figures but would not fundamentally change the conclusion of the current work; in fact, it may even take away from the targeted integration of molecular biology and neuroscience we have tried to achieve.

Beyond this, we do not have such an unallocated budget, and so this new project would require us first to generate preliminary data on the bitter neurons to write then a grant proposal to fund it; as you can appreciate, this would take longer than a year, especially since we do not even know if the bitter gustatory neurons are affected by a high-sugar diet. Beyond this, looking at the bitter neurons would do little to prove specificity. If we found no effects of this pathway on the activity of the bitter neurons, it wouldn’t establish that the changes in the sweet taste neurons are specific. In fact, the same pathway could be acting in some of the other thousands of fly circuits that were not investigated (Black swan effect). If we did find that OGT/PRC2/Sr play a role in the bitter neurons, it would also do little to disprove specificity since their targets would likely be different because the sets of genes expressed in these two sensory neurons are different. By analogy, the protein sensor mTOR is expressed and active in every cell, where it modulates some of the same targets (i.e., S6K); however, the effects of the pathway may be different due to the distinct metabolic and genetic idiosyncrasies of cells, as well as cellular compartments. This lack of specificity doesn’t mean that mTOR is not important. Finally, we would like to note that we have tested the effects of manipulating OGT levels in other neurons (dopamine and Mushroom Body Output Neurons) without effects on behavior or neural responses (May et al. 2020; Pardo-Garcia et al. 2022); based on these, OGT doesn’t seem to affect neurons indiscriminately.

Budget ~ $129,000

Salary and fringe benefits for PD for 10 calendar months: $75,000 (2 months behavior experiments, 2 months training for molecular biology experiments and troubleshooting in new neurons, 4 months growing flies and conducting experiments, 2 months data analysis and visualization)

DAM ID: $16,0000, Pcl:dam and OGT:dam in CD and SD, with and without OSMI x 4 biological replicates per condition = 32 samples @ $500 per sample (UM Genomics core)

TRAP: $16,0000 Pcl mutant and OSMI in CD and SD x 4 biological replicates per condition + sequencing input = 32 samples @ $500 per sample (UM Genomics core)

Animals' food and maintenance: $ 5,000, $500 per person/10 months

Reagents: = $17,000, including sequencing kit (32 reactions = $6,000) x 2 = $12,000, and other reagents such as drugs and plastic

Note that this PD would have to be hired and retrained. The first author of the manuscript who carried out the molecular experiments graduated in Dec 2021 but failed to pass on the technical knowledge due to COVID restrictions at the UM: we were completely shut down until July 2020, and at 20% capacity from March 2020 to July 2021 (people couldn’t also work together to show techniques), and no new people joined the lab in 2020-2022 (most of the 2021 grad student class deferred to 2022 and no PD candidates apply).

7) The Proboscis extension response needs to be further elaborated on. The data plotted seems to be normalized but this is not explained in the paper. What is actually being measure? What is the N? If the data is normalized, then the authors should present raw data graphed in a supplemental figure at least.

The data shown are the raw data plotted as mean +/- SEM. In the proboscis extension response, what is measured is the extent of the proboscis extension when the fly labellum is stimulated with different sucrose concentrations. The extent of proboscis extension is measured as 1 = full extension, 0.5=half extension, and 0=no extension. We added this as a diagram in Figure 3A.

8) Figure 4B seems misleading in the way it is plotted. Authors claim a bias of SR genes at the TSS in the PRC2.1 and OGT overlapping regions. However, the y axes are not to the same scale. In fact, the not OGT/PRC2 regions have much higher number of genes with SR motifs at every single position plotted on 4B. Specifically the strong statements findings in the results paragraph on lines 262-283 are not sufficiently supported and the authors are encouraged to revise this or move it to discussion.

4B (Now 5B) plots the distribution of Sr binding sites along the regions up and downstream of the TSS. The reason for the difference in the Y axis is that far more genes have Sr binding sites and do not have OGT/PRC2/Pcl. What the graph clearly shows, however, is where the position of Sr is relative to the TSS, and the pattern is opposite between OGT/Pcl and non-OGT-Pcl. To address the reviewer’s comment, we have now normalized the counts relative to the genome-wide expectation (derived by multiplying the number of potential target genes by the fractional coverage of Sr motifs on the genome); a score of 1.0 indicates the hypothetical genome-wide average overlap with Sr motifs and is shown as a red dashed line. This normalized analysis shows an even stronger enrichment of Sr at the TSS of OGT/PRC2/Pcl genes.

9) Many of the experiments involved moving flies to OSMI or Trametinib diets. A caveat to this method is that this treatment will influence OGT/ERK in the fly globally. Because both OGT/ERK impact a number of critical biological processes, this makes the results difficult to interpret as impacts of taste sensing or due to the flies being generally unhealthy. It would be helpful to see fly survival on control, OSMI supplemented, and Trametinib diets to determine how detrimental these diet conditions are in general to the fly.

We have now added these experiments to Figure 4-Supplement 1A. There is no effect on survival. Also, OSMI and Trametinib affect the taste system in specific dietary contexts, not indiscriminately.

10) In general, the figure legends and methods are lacking in detail. For instance, in Figure 1F, are the GO terms for control diet comprised from all peaks listed or just the peaks unique to CD? Same question for the SD GO terms. Abbreviations are not elaborated in the legend or methods (ex: Figure 2B TPM) or lacking axis labels (ex: Figure 2D), etc. I suggest a thorough review of the methods and legend to add details necessary to interpret the results and make it possible to replicate experimental designs.

In Figure 1, “CD” means the Control diet only GO terms, “SD” means the Sugar diet only go terms, and SD/CD the differential binding go terms. We further clarified this in the legends and text and added the missing abbreviation.

Reviewer #2 (Recommendations for the authors):1. There are too many instances of non-technical phrases that are unclear/nonspecific. For example, "…OGT moonlights on the chromatin ….", or "… OGT decorates nutrient-sensitive …". These phrases should be replaced with more precise words, unless these phrases are common in the field and I am just unaware.

We were also surprised to see these terms used in the literature. However, “moonlight” is indeed generally used to describe alternative functions of enzymes, especially metabolic enzymes. The term was generated early in the 2000s, but it has been used more in the last 5-10 years; currently, there are 1,700 manuscripts on Pubmed that use “moonlight” as we did in the manuscript. “Decorate” first originated to describe post-translational or non-enzymatic modifications to histones, but more recently, it has also been used to refer to the association of a protein with chromatin or nucleic acids; this is widely used in the epigenetics field, perhaps because some find that using “binding” or “association” may imply direct interactions with DNA or chromatin. Here are some recent examples:

Chaoyun Pan, Bo Li, M Celeste Simon. Moonlighting functions of metabolic enzymes and metabolites in cancer. Mol Cell. 2021 Sep 16;81(18):3760-3774. doi: 10.1016/j.molcel.2021.08.031.

Boukouris AE, Zervopoulos SD, Michelakis ED. Metabolic Enzymes Moonlighting in the Nucleus: Metabolic Regulation of Gene Transcription. Trends in Biochemical Sciences, 23 Jun 2016, 41(8):712-730 DOI: 10.1016/j.tibs.2016.05.013

Youjun Zhang, Arun Sampathkumar, Sandra Mae-Lin Kerber, Corné Swart, Carsten Hille, Kumar Seerangan, Alexander Graf, Lee Sweetlove & Alisdair R. Fernie. A moonlighting role for enzymes of glycolysis in the colocalization of mitochondria and chloroplasts. Nature Communications volume. 11, Article number: 4509 (2020)

From Arnould C, Rocher V, Finoux AL, Clouaire T, Li K, Zhou F, Caron P, Mangeot PE, Ricci EP, Mourad R, Haber JE, Noordermeer D, Legube G. Loop extrusion as a mechanism for formation of DNA damage repair foci. Nature. 2021 Feb;590(7847):660-665. “When a DSB forms, the PI3K-related ATM kinase rapidly triggers the establishment of megabase-sized, chromatin domains decorated with phosphorylated histone H2AX (gammaH2AX), which act as seeds for the formation of DNA-damage response foci.”

Chul-Hwan Lee, Jia-Ray Yu,1,7 Sunil Kumar, Ying Jin,3 Gary LeRoy, Natarajan Bhanu, Syuzo Kaneko, Benjamin A. Garcia, Andrew D. Hamilton, and Danny Reinberg. Allosteric activation dictates PRC2 activity independent of its recruitment to chromatin. Mol Cell. 2018 May 3; 70(3): 422–434.e6. “Mammalian heterochromatin contains large repressive chromatin domains, including H3K9me2/3-decorated constitutive heterchromatin at pericentromeric and telomeric regions.”

2. The Discussion section is overall too long and delves too much into the previous literature. It should be condensed and there should be more focus on the translational implications of the results.

We thank the reviewer for the comment; we have edited and shortened the discussion.

Reviewer #3 (Recommendations for the authors):– Almost all quantitative data in figures lack definition of variance measures (e.g. Panel 3C and E). Are data represented as mean +/I standard deviation, s.e.m.?

We wrote this in the methods but not in each legend; we apologize for the omission and have now added this to each legend.

– Regarding setting up the problem – claiming "how nutritional information is transmuted to neural activity and behavioral changes remains elusive" is inaccurate – we know the receptors, increasingly many cells in the food intake regulatory circuit from sensory input to motor output in *Drosophila*.

It was not our intention to downplay previous findings; we used nutritional information to refer to metabolic signaling, but the reviewer's point is correct that the circuitry and molecular perception of sugar are well-established and characterized in flies and thus responses to nutritional information. We have now edited this sentence to make it clearer to what we wanted to refer to.

– Methods refer to a python code on Github but no github repository is specified.

We apologize for the omission; we have added the python script to the supplementary file 2.

– Please also make the code for analyzing the DaMID and CaTaDa available online.

The protocol and code we used to analyze these data were published in (Marshall and Brand 2015). We did not generate a new analysis code.

References

Mariappa, Daniel, Xiaowei Zheng, Marianne Schimpl, Olawale Raimi, Andrew T. Ferenbach,H-Arno J. Müller, and Daan M. F. van Aalten. 2015. “Dual Functionality of O-GlcNAc Transferase Is Required for *Drosophila* Development.” Open Biology 5 (12): 150234.

Marshall, Owen J., and Andrea H. Brand. 2015. “Damidseq_pipeline: An Automated Pipeline for Processing DamID Sequencing Datasets.” *Bioinformatics* 31 (20): 3371–73.

May, Christina E., Julia Rosander, Jennifer Gottfried, Evan Dennis, and Monica Dus. 2020. “Dietary Sugar Inhibits Satiation by Decreasing the Central Processing of Sweet Taste.” *eLife* 9 (June). https://doi.org/10.7554/*eLife*.54530.

Ortiz-Meoz, Rodrigo F., Jiaoyang Jiang, Michael B. Lazarus, Marina Orman, John Janetzko, Chenguang Fan, Damien Y. Duveau, Zhi-Wei Tan, Craig J. Thomas, and Suzanne Walker.

2015. “A Small Molecule That Inhibits OGT Activity in Cells.” *ACS Chemical Biology* 10 (6): 1392–97.

Vaziri, Anoumid, Morteza Khabiri, Brendan T. Genaw, Christina E. May, Peter L. Freddolino, and Monica Dus. 2020. “Persistent Epigenetic Reprogramming of Sweet Taste by Diet.” *Science Advances* 6 (46). https://doi.org/10.1126/sciadv.abc8492.